# Performance Analysis of IoT-Based Health and Environment WSN Deployment

**DOI:** 10.3390/s20205923

**Published:** 2020-10-20

**Authors:** Maryam Shakeri, Abolghasem Sadeghi-Niaraki, Soo-Mi Choi, S. M. Riazul Islam

**Affiliations:** 1Geoinformation Technology Center of Excellence, Faculty of Geodesy & Geomatics Engineering, K.N. Toosi University of Technology, Tehran 19697, Iran; mshakeri@mail.kntu.ac.ir; 2Department of Computer Science and Engineering, Sejong University, Seoul 05006, Korea; smchoi@sejong.ac.kr (S.-M.C.); riaz@sejong.ac.kr (S.M.R.I.)

**Keywords:** wireless sensor network deployment, health and environment applications, IoT, Bees Algorithm, PSO algorithm, coverage, lifetime, Minimum Spanning Tree

## Abstract

With the development of Internet of Things (IoT) applications, applying the potential and benefits of IoT technology in the health and environment services is increasing to improve the service quality using sensors and devices. This paper aims to apply GIS-based optimization algorithms for optimizing IoT-based network deployment through the use of wireless sensor networks (WSNs) and smart connected sensors for environmental and health applications. First, the WSN deployment research studies in health and environment applications are reviewed including fire monitoring, precise agriculture, telemonitoring, smart home, and hospital. Second, the WSN deployment process is modeled to optimize two conflict objectives, coverage and lifetime, by applying Minimum Spanning Tree (MST) routing protocol with minimum total network lengths. Third, the performance of the Bees Algorithm (BA) and Particle Swarm Optimization (PSO) algorithms are compared for the evaluation of GIS-based WSN deployment in health and environment applications. The algorithms were compared using convergence rate, constancy repeatability, and modeling complexity criteria. The results showed that the PSO algorithm converged to higher values of objective functions gradually while BA found better fitness values and was faster in the first iterations. The levels of stability and repeatability were high with 0.0150 of standard deviation for PSO and 0.0375 for BA. The PSO also had lower complexity than BA. Therefore, the PSO algorithm obtained better performance for IoT-based sensor network deployment.

## 1. Introduction

Internet of Things (IoT) has changed operational health and environment systems with sensors and devices to turn a stream of right information into some sort of intelligence location based service [1,2,3,4]. It has to be highlighted that the Wireless Sensor Network (WSN) plays a significant role in IoT because it can be developed in many fields [5,6], so that the most cities have deployed or are planning to deploy WSN for creating city-wide wireless coverage [7]. In environmental and health monitoring applications, the WSNs have to be designed extremely reliably because it can be literally the difference between life or death. It means that they have to be designed in such a way that they ubiquitously provide the environmental and health services at any place and at any time. To deploy such a WSN, due to the high cost of the sensors, the minimum number of sensors has to be located in the position to create maximum coverage of the region. On the other hand, the batteries of the sensors limit the provision of the health and environmental services ubiquitously. This issue creates more challenges for WSN deployment in the environment and health applications due to the difficult access areas in the environment (e.g., mountains, forests, and groundwater) as well as the sensitivity areas in the hospitals where it is not possible to replace the sensors’ batteries continuously. To consider this issue, WSN has to be deployed so that the sensors consume less energy. Therefore, coverage and energy consumption are two important issues in the WSN deployment in environmental and health applications.

To design a reliable network, the topology of the network is very important, which refers to the arrangement of different elements of an IoT network by choosing the number and location of the sensors, communication protocols, packet size, and radio range [1,8]. Topology is one of the elements of the architecture of IoT network that controls the network to provide environmental and health services in seamless environments [1]. In this regard, the locations of the sensors plays significant role to design proper topology of the network and highly correlated to its reliability, security, and safety [8,9,10,11,12]. Determining the locations of the sensors in the network that is defined as a deployment problem of the WSN based environmental and health system can be done using Geospatial Information Systems (GIS) and be useful for effective environmental and health planning [13,14,15]. WSN deployment is a multi-objective optimization problem to find optimal locations of sensor nodes in the network to provide seamless environmental and health services with a trade-off between conflict objectives [16,17] including lifetime and coverage. Based on [18], multi-criteria decision making (MCDM) and evolutionary multi-objective optimization (EMO) algorithms are applied to solve multi-objective optimization problems. This research focuses on EMO algorithms that have widely utilized for solving multi-objective optimization problems in different fields.

There have been few studies to design WSN in IoT by considering different objectives of the network deployment for environmental and health monitoring applications. Abdel-Basset et al. [19] designed a WSN for IoT smart agriculture to maximize the area coverage of using a novel metaheuristic algorithm, a multi-verse optimizer. Zheng et al. [20] deployed WSN for earth temperature monitoring to optimize the network topology and to reduce energy consumption. Sudha et al. [21] designed a WSN based network to monitor physiological parameters of patients that reduce the energy consumption to prolong the network lifetime and extend the communication coverage. Fariborzi et al. [22] proposed an energy-aware architecture of WSN for ubiquitous healthcare systems for hospitals to increase the lifetime of the network and cover the area of the hospital. Varshney [23] presented requirements of pervasive healthcare using WSN including coverage and scalability. Although these research addresses coverage and lifetime in the deployment of the sensor networks, this research optimize both coverage and lifetime together for the WSN deployment in environmental and health application using artificial optimization algorithms. In this regard, several research studies have been conducted to optimize both coverage and lifetime in general WSN deployment [16,17,24,25,26]. Although such research has addressed both important issues in the sensor network deployment, these studies haven’t compared two evolutionary optimization algorithms (Bees Algorithm (BA) and Particle Swarm Optimization (PSO) algorithms) for WSN deployment with considering coverage, lifetime and connectivity objectives. In addition, they haven’t used a Minimum Spanning Tree (MST) algorithm for path and energy optimization in WSN deployment.

This paper aims to apply GIS-based optimization algorithms for WSN deployment in the health and environmental applications for maximizing the coverage and lifetime by considering the connectivity constraint. To achieve this main goal, three sections of research are as follows. First, previous research studies on WSN deployment are reviewed in environmental and health applications. Second, the MST algorithm has been applied in the WSN deployment process to minimize energy consumption. MST algorithm is a routing protocol that minimizes WSN energy consumption with minimum total network lengths. Third, two optimization algorithms, BA and PSO, are compared for evaluating the GIS-based WSN deployment in health and environmental applications. BA and PSO are the evolutionary optimization algorithms that have been successfully applied for solving different multi-objective problems. To compare the algorithms, three parameters are used: convergence rate, constancy repeatability, and modeling complexity.

Section 2 reviews WSN deployment research studies in several applications of environmental and health applications and shows the importance of coverage and lifetime in WSN deployment for these fields. Section 3 describes the methodology of the research. This section includes the GIS-based WSN deployment method for optimizing the network coverage and lifetime with applying the MST routing algorithm. In addition, the steps of BA and PSO optimization algorithms for GIS-based WSN deployment are explained in this section. Section 4 covers how the algorithms implemented in a simulated health and environmental region and the experimental results of comparing two algorithms using three criteria. Section 5 belongs to the discussion of the research. Finally, conclusions and future works are explained in Section 6.

## 2. Literature Review

WSN deployment has been needed in different health and environmental applications. Some WSN deployment applications for the environment are water, air pollution, flood, agriculture, earthquake, landslide, habitat, soil, temperature, and fire monitoring in forests and urban areas. In the water domain, Pule et al. [27] by reviewing WSNs on monitoring water quality, demonstrated the importance of coverage and energy consumption and security of WSN in most of the studies. Saini [28] who reviewed the WSN research about water quality pointed out the challenges of WSN in monitoring water quality including the low energy of sensors, connectivity, and sensor location. Yao and Du [29] maximized underwater coverage for WSN deployment in a 3D environment using a gradient direction algorithm. Ellouze et al. [30] proposed an algorithm to deploy a set of fixed RFID tags for detection and localization of polluted areas throughout waterways by minimizing energy consumption and the number of sensors. In air pollution monitoring, Boubrima [31] and Boubrima et al. [32] confirmed that the precise WSN deployment considering lifetime and coverage is necessary for both applications of air pollution monitoring: generating maps, and detecting high polluted regions. They presented an integer linear programming modeling for the WSN deployment with coverage, connectivity, lifetime, and cost constraints. In flood prediction and prevention, Priyadarshinee et al. [33] reviewing the applications and key factors of the WSN, referred to the coverage and energy consumption as the important criteria in the WSN deployment.

In agriculture, as Rajput and Kumaravelu [34] stated, the scalability, coverage, and lifetime prolongation are the main requirements of WSN deployment. Abdel-Basset et al. [19] optimized the area coverage of WSN for IoT smart agriculture using a novel metaheuristic algorithm. Sharma et al. [35] proposed a framework for enhancing WSN lifetime using solar energy harvesting to measure plant watering, the moisture level of soil and plants, crop harvesting, pesticide distribution, and animal controlling in smart agriculture monitoring. Rajput and Kumaravelu [36] developed a Fuzzy c-means algorithm to reduce data transmission distances and as a result to optimize the network energy efficiency and coverage in agriculture monitoring. In the earthquake domain, Dutta et al. [37] proposed a model for quick WSN deployment by considering gradual changes to environmental characteristics to determine the seismic event magnitude and distance to seismic sources. Hung et al. [38] developed an on-site earthquake early warning system using energy-aware WSN to reduce power consumption. Mohapatra et al. [39] proposed the network of ocean-bottom seismometer sensors for real-time monitoring using combined routing and node replacement approaches to reduce energy consumption and minimize the costs of replacement the sensors. In landslide monitoring, Kumar et al. [40] dealt with a large-scale IoT based WSN deployment problem for remote monitoring landslide prone zones and reporting real-time measurements by considering network connectivity, coverage, the optimal number of nodes, and time synchronization. Giorgetti et al. [41] proposed the WSN deployment process for online analysis and the alerting of landslide supervision to evaluate the risks and provide useful information in Torgiovannetto (Italy) by prolonging the network lifetime. Ramesh [42] described design, development, and deployment WSN at Anthoniar colony, Munnar, Idukki (Dist), and Kerala (State) in India for landslide detection by considering the power consumption of the network.

In habitat monitoring, deploying audio sensors with limited energetic and memory capacity to monitor bird sounds have been addressed in Boulmaiz et al. [43]. Naumowicz et al. [44] presented the WSN deployment on Skomer Island to monitor spatial behaviors of seabirds to get and assess information about the health of the ecosystems. Naumowicz et al. [45] described the design and deployment of WSN in the UK National Nature Reserve for seabird monitoring. In temperature monitoring, Zheng et al. [20] designed a ZigBee technology based WSN for earth temperature monitoring. To design WSN, the network topology and routing protocol have been optimized to reduce energy consumption. Baghyalakshmi et al. [46] designed and developed a WSN based monitoring system for temperature and also humidity monitoring for the open terrace and control room in the SADHANA facility. Yang et al. [47] addressed WSN topology for remote temperature monitoring using clustering to reduce energy consumption. In the fire monitoring and the construction of fire rescue to prevent economic loss and environmental damage, WSN placement in the environment is done to determine and monitor the location of trapped [48]. Al-Turjman et al. [49] emphasized the importance of WSN forestry deployment with considering lifetime, coverage, and energy consumption as a future work. Li et al. [50] designed a WSN for wildfire detection and alarm signaling that maximizes robustness and network lifetime using TinyOS kernels to implement localization and power management. Lei and Lu [51] developed a computational algorithm for the WSN placement to build near real-time systems for forest fire detection with optimizing complete coverage of the sensor nodes and network lifetime. Abdulsahib and Khalaf [52] divided WSN in randomly-spread nodes to improve the network coverage and energy consumption. Biabani et al. [53] proposed the hybrid Particle Swarm Optimization (PSO) and Harmony Search Algorithm (HSA) algorithm for clustering and routing in WSN deployment by considering the network energy, closeness and coverage criteria in forest fire management.

In health monitoring and healthcare system, different applications need a careful designing and deploying of WSNs to improve and expand the quality of care [54]. Deploying WSNs for telemonitoring [55] in smart cities is one of these applications. In telemonitoring, WSNs should be deployed to connect the hospitals, medical centers, and care institutions by considering city coverage, connectivity, and reducing energy consumption so that the health medical services are immediately available throughout the city. In the hospital domain, Fariborzi et al. [22] developed a ubiquitous health network for hospitals and care institutions using three types of sensor nodes (patients, routers a coordinator). They deployed in such a way that the sensors located in different locations of the hospitals to cover the area of the hospital and reduce power consumption to provide real-time and non-intrusive monitoring of the patients’ vital signs. In addition, deploying an IoT based WSN system to manage the medical devices (blood pressure monitor, wheelchairs and etc.) and medicines is an important application of WSN [56,57,58,59]. The WSN must continuously track the equipment and medicines to help clinical staff, doctors and nurses to quickly and immediately find them in real-time [60]. One of the applications is generating the network of ambulances using IoT based WSN systems to effectively respond in emergency situations [61,62,63]. This WSN should be deployed to cover the whole of the smart city so that when an accident occurs in every place of the city, emergency medical services could be offered at the right time. In addition to the coverage, the communication between the sensors is necessary to reduce response time. Deploying a WSN for monitory the health of a person in the home in perform daily activities is another application of WSN. Suryadevara et al. [64] addressed the deployment of WSN to monitor household objects, personnel movements, environmental parameters, and human physiological parameters for elderly persons in smart homes. In deploying the network of heterogeneous sensors, they considered tree-based wireless communication topology to manage energy consumption. Liu et al. [65] developed a genetic algorithm (GA), particle swarm optimization (PSO), and ant colony (AC) to maximize the coverage of the WSN in three-dimension (3D) region to monitor chronic patients in two scenarios: nursing house or smart home.

## 3. Health and Environment Sensor Network Deployment

Based on what can be inferred from the previous research in several applications of health and environment (Section 2), the sensor network deployment is a multi-objective optimization problem to maximize coverage and lifetime with considering the connectivity constraint. To solve this multi-objective optimization problem, GIS based optimization algorithms are applied in this research. The methodology of applying and evaluating GIS based optimization algorithms to optimize coverage and lifetime in health and environmental applications are shown in Figure 1. In the first step, GIS based optimization is modeled for the WSN deployment. In this regard, coverage and lifetime functions are presented to calculate the total coverage and lifetime of the network. The MST routing protocol is applied to generate a unit distance graph of the sensors to optimize the topology and then minimize the energy consumption. Second, the BA optimization algorithm is designed to step by step deploy health and environment sensors. BA is one of the artificial algorithms that has been most extensively used in many real-world applications [66] and it is an efficient algorithm that needs fewer control parameters and easy to implement [67,68]. Third, the PSO algorithm procedure is designed to solve GIS based multi-objective WSN deployment. PSO is one of the swarm intelligence algorithms, which simulates the behavior of bird flocks and fish schools [69,70]. Fourth, the BA and PSO algorithms are implemented for a simulated region. Before testing the algorithms, the algorithm parameters are set using the parameter tuning method. Finally, two BA and PSO algorithms are compared with three criteria.

### 3.1. GIS-Based WSN Deployment

As the previous section shows, WSN deployment is one of the important problems in several environmental and health applications with optimizing conflict objectives (coverage and power consumption or lifetime). The aim of WSN deployment is to find the optimal locations of the sensors so that the network coverage and lifetime are maximized. In WSN deployment, GIS plays key role in modeling the sensors that are located in a specific location (*x*, *y*) using point geometry and managing the network of the sensors. In addition, spatial analysis of GIS helps to calculate total coverage of the sensors as well as the overlapped coverage in the given region. The locations of the sensors have great impact on the WSN coverage and energy consumption and even cost. If the locations of the sensors is such that they have the maximum distance from each other according to the sensing range (the points have uniform spatial distribution), the network coverage is maximum and the number of sensors is minimum for the WSN deployment. However, increasing the distance between the sensors increases energy consumption. To reduce energy consumption, it is necessary to keep the distance between the sensors as short as possible, which this issue increases the number of sensors and thus the cost. Therefore, the main issue of the WSN deployment is to find the optimal locations of the sensors by considering both the sensing and communication ranges. In addition, to find optimal locations, the connectivity constraint is very important so that the sensors that are not located in the communication range cannot be connected and to make a network.

WSNs, as a specialized type of wireless sensor network (WSN) in geospatial spaces, are used for computing everywhere. A WSN is a collection of tiny battery-powered sensor nodes in which information generated by each sensor node is directly related to its location [71]. A WSN can be considered as connected undirected graph represented by *G* (*V*, *E*), where *V* = {$v_{1},\;v_{2},\;v_{3}\ldots ,\;v_{n}$} is a finite set of vertices representing sensor nodes, and *E* = {$e_{1,2},\;e_{1,3},\;\ldots ,\;e_{i,j},\;\ldots ,\;e_{n-1,\;n}$} is the edges representing the connection links between sensor nodes [72]:

WSN, a set of spatially distributed sensor nodes, is deployed in a geospatial space to detect and monitor different phenomena [73,74,75]. The robustness of WSN in collecting and transmitting data is affected by the sensor limitations including sensing range, communication, power supply, and computational capacities. In WSN deployment, as one of the fundamental and major steps of the design process, finding sensor positions are conducted with regard to reducing the effects of sensor limitations. Therefore, WSN deployment is inherently a multi-objective problem. Among the different objectives, coverage and lifetime of the network are two main important objectives for increasing the effectiveness of WSNs [16,17]. These objectives are limited by the connectivity constraint that there should be at least one link between sensor nodes [76].

Each sensor has several properties including sensing range and communication range. The sensing range assumes a circle with radius $R_{s}$ in which the sensor is able to sense all the points. The communication range is Euclidean distance $R_{c}$ that two sensors are able to communicate with each other if their distance is less than or equal to their communication ranges [16]. These ranges are shown in Figure 2. Each edge has m attributes denoted by $w_{i,j}=\left\{w_{i,j}^{1},\;w_{i,j}^{2},\;\ldots ,\;w_{i,j}^{m}\right\}$ to represent distance, transmitting energy and so on. In addition, the connectivity between two nodes (node *i* and *j*) is defined with $x=x_{1,2},\;x_{1,3},\;\ldots ,\;x_{i,j},\;\ldots ,\;x_{n-1,\;n}$. The value of $x_{i,j}$ is one if there is an edge ($e_{i,j}$) between the nodes and the edge is selected, otherwise zero.

As previous research in the environmental and health applications confirmed (such as [28,29,51,65]), WSN deployment affects the coverage and communication among sensor nodes in the network. This process which is the determination of the locations of sensor nodes is conducted with optimizing certain multi-objectives according to the sensor properties [16]. The multi-objective optimization can be formulated as a linear function f consisting of m objectives to be maximized for the problem with considering connectivity constraint. The objective (fitness) function can be formulated as:(1)$$\max f\left(x\right)=\;\alpha \;f_{1}\left(x\right)+\;\beta \;f_{1}\left(x\right)+\ldots +\;\gamma \;f_{1}\left(x\right)+\;\zeta \;f_{1}\left(x\right)$$
(2)$$\max f_{k}\left(x\right)=\;\sum_{j=1}^{n}w_{i,j}^{k}x_{i,j},\;i=\;j-\;1$$
where $f_{i}\left(x\right)$ is the objective function to be maximized for the problem by considering α+β+…+ γ=1.

#### 3.1.1. Coverage

The coverage which is the fundamental issue in WSNs addresses how to deploy sensor nodes in the geographic space so that every point of the space is sensed at least by one sensor node [16,73,77]. This is an optimization problem to find optimal locations of sensor nodes in the network so that the coverage of the space is maximized [17,73]. This issue is addressed as the network coverage in the WSN deployment which is calculated using the sensor coverage. The sensor coverage is defined as the area of the $R_{s}$-meter circle, where $R_{s}$ is the sensing range of the sensor [78]. The total coverage of WSN, which is shown in grey colour in Figure 2, is calculated as follows:(3)$$\mathrm{coverage}=\overset{n}{\underset{i=1}{\sum }}(C_{i}-C_{i}^{o})/C_{R}$$
where $C_{i}\;$is the coverage of the ith sensor, $C_{i}^{o}$ is the overlapped coverage between the ith sensor and other sensors, $C_{R}\;$is the total area of the region of interest (ROI). In this formula, $C_{R}$ is used to normalize the total coverage. So, the total coverage is a normalized value between 0 and 1. The total coverage of the sensor network should be maximized in the design process. By assuming that the existing sensors can rotate 0–360° horizontally [73], each sensor coverage and total network coverage are calculated by spatial analyses. To this end, first, a point geometry layer is created using the locations of the sensors in which a point represents a sensor node. Then, the buffer polygon around the points are created to display the coverage area of the sensor in the sensing distance using buffer analysis. After that, the overlapped area between two polygons are removed from using Intersection and Union analysis. Buffer, Intersection, and Union are the spatial analyses that are performed on different types of the geometry in GIS [79]. 

#### 3.1.2. Lifetime

Energy power consumption of battery-powered sensor nodes of WSN especially for transitions of information is one of the constraints of the network that affect the lifetime of the network and thereby the applicability for environmental and health purposes. Lifetime is one of the big challenges of adopting WSN for environmental and health services [5]. Lifetime addresses how to reduce the energy consumption of the sensor nodes by increasing the time until one of the sensor nodes runs out of energy [16,67]. Therefore, It is the time until the energy supply of the first sensor node is depleted, which is the maximum number of sensing cycles. The coverage and lifetime are affect to each other so that obtaining a good coverage objective is related to network lifetime or energy saving [66]. As energy is a vital resource in designing WSNs, lifetime is considered the other main challenge in much research [16,17,80]. Lifetime can be stated as [16]:(4)$$\mathrm{lifetime}=\frac{\min\;{\left\{T_{\mathrm{failure},\;i}\right\}}_{\;i=1,\;2,\;3,\ldots ,n}}{T_{\max}}$$
(5)$$T_{\max}=\frac{E\;\left(\mathrm{initial}\;\mathrm{energy}\right)}{E_{\min}*\in_{\mathrm{amp}}}$$

Only the transmitting ($E_{T}$) and receiving energy ($E_{R}$) are considered for WSN lifetime, which is calculated as follows for one unit message at a distance of *d* [81]:(6)$$E={\;E}_{T}\;+\;r{\;E}_{R}$$
(7)$$E_{T}=(E_{elec}+\;\in_{\mathrm{amp}}\;d^{2})$$
(8)$$E_{R}=\;r\;E_{elec}$$

In the equations, *r*, $E_{elec}$, and $\in_{\mathrm{amp}}$ represent the number of receiving, the transmitter or receiver circuitry, and the transmit amplifier, respectively. The number of receiving is used to calculate the total energy consumptions from the viewpoint of receiving messages by Equation (12) based on the [49], because the sensors consume energy power to receive each unit message. The number of reviving messages is calculated using the number of inbound links of each sensor node. Since coverage and lifetime are important issues in WSN deployment, the objective function can be formulated as follows:(9)$$\max f\left(x\right)=\;\alpha \left(\mathrm{coverage}\right)+\beta \left(\mathrm{lifetime}\right)=\alpha \frac{\sum_{i=1}^{n}C_{i}-C_{o}}{C_{R}}+\beta \;\frac{\min\;{\left\{T_{\mathrm{failure},\;i}\right\}}_{\;i=1,\;2,\;3,\ldots ,n}}{T_{\max}\;}$$

#### 3.1.3. Routing Algorithm

To manage the energy power consumption for maximizing the lifetime of the network, a routing protocol is required to consider in the designed network [5,80]. There are many different routing protocols, however, Minimum Spanning Tree (MST) is one of them to optimize the energy consumption by minimizing the total transmission [82]. MST is a tree where any pair of vertices are connected by exactly one path and is a unique spanning tree with a minimum total edge length [71]. Minimum spanning tree is a weighted subgraph of spanning tree which is a subgraph of an undirected and connected graph without loops [83]. In general, a tree is a spanning subgraph of unit distance graph (UDG). The UDG is formed by creating edges to connect all nodes that are within the communication range (called unit distance), $R_{c}$. UDG can formally be defined as [71]:(10)$$\mathrm{UDG}=\left(V,\;\left\{\left(u,v\right)|\left(u,v\right)\in E\;\mathrm{and}\;0<\delta \left(p\left(u\right),\;p\left(v\right)\right)<R_{c}\right\}\right)$$
(11)$$\delta \left(p\left(u\right),\;p\left(v\right)\right)=\;\sqrt{\Delta X_{u,v}{}^{2}+\Delta Y_{u,v}{}^{2}}$$
where *p* is the position of sensor nodes in $R^{2}$. MST that connects all *n* sensor nodes with *n* − 1 edges has minimum total weights of edges [71,80]. In MST, the length of each of the edges can be considered as its weight; in this case, the sum of the length of MST edges is minimum. There are two efficient algorithms to construct MST including Kruskal [84] and Prim [85] algorithms.

### 3.2. BA-Based Sensor Network Deployment

In the BA algorithms, the optimized locations of sensor nodes of the network are determined by simulating bees’ behaviors in seeking the best food source with the aim of maximizing the coverage and lifetime. The BA contains three kinds of bee including employed bees (assigned to food sources), onlooker bees (watching the dance of employed bee), and scout bees (searching new food sources randomly) [66]. Figure 3 shows the BA-based sensor network deployment algorithm. The algorithm is explained step by step below.

Step 1: Generate an initial random population with *p* number of employed bees that creates a bee colony. Each bee, representing one sensor network (Figure 4), is generated by UDG with *n* vertices at random position. Each vertex represents a sensor node, a resource, in the WSN. The vertices that are not located in the communication range of the sensors are not connected to each other in the graph. The connectivity of the generated random graph is checked by using its adjacency matrix (G) if and only if [16].
(12)$$Y=G+G^{2}\ldots +{\;G}^{n-1}\;\neq 0$$
(13)$$G={\left\{x_{i,j}\right\}}_{n\times n}\;i=1,\;\ldots ,\;n;j=1,\;\ldots ,\;n$$If the generated random graph is not connected, the positions of some vertices are changed until the connected UDG graph is obtained. To do this, first the connected components of the graph are found, then the positions of the vertices of a component with the minimum number of the vertices are changed toward the nearest component.Step 2: Generate MST based on the Kruskal algorithm and distances between the sensor nodes as weights of the edges from the generated UDG.Step 3: Evaluate fitness function, according to Equation (13), of the employed bees of the initial population.Step 4: Set t = 0. The following steps are done until t less than the number of generations (t < $N_{\mathrm{iteration}})$.Step 4.1: Select m bees that have higher fitness value ($f_{i}$) as better bees.Step 4.2: Select e elite bees from m selected bees.Step 4.3: Recruit onlooker bees to search in the neighborhood of each selected e elite and (*m - e*) better bees and evaluate their fitness. To do this, the neighborhood search distance and size for searching around each type of selected bee. The neighborhood search distance shows how many vertices of the bee can be changed randomly in the sensing range. In other words, the searching distance in the sensor network deployment is a number that determines the position of maximum how many sensors can be changed based on the type of bees (dep for elite bees and dbp for better bees). Therefore, in this step, the position of the random number of vertices is changed randomly in the neighborhood search of the bee, as below:
(14)$$x_{i}={\;x}_{i}\;\pm \mathrm{Rand}\;\left(0,\;\;d\right)$$
(15)$$y_{i}={\;y}_{i}\;\pm \mathrm{Rand}\;\left(0,\;\;d\right)$$The searching size means the number of onlooker bees for searching around the bees. The number of onlooker bees in the WSN deployment is the number of new bees that can be generated by changing the position of the network vertices of the selected bees. The number of onlooker bees for elite and better bees is different, so nep and nbp are the numbers of onlooker bees sent to search the neighborhood around the elite and better bees, respectively. It should be noted, the network connectivity also is checked after the position of the vertices changed.Step 4.4: Select a best bee at each neighborhood search.Step 4.5: Assign (s = n − m) scout bees to randomly search and evaluate their fitness.Step 4.6: t = t + 1.Step 5: Find the best global bee (WSN).

### 3.3. PSO-Based Sensor Network Deployment

In a PSO algorithm, the deployment problem of the sensor network is simulated as swarm particles, which move around the search space and are improved by their own best position and swarm’s best position together through generations to find the optimum solution [68,69]. Figure 5 shows the PSO steps for MST-based network deployment algorithm to maximize coverage and lifetime. The algorithm is explained step by step below.

Step 1: Generate an initial random population with *p* number of particles representing sensor networks. This step is completely equivalent to that of BA. In this algorithm, the position of the resources is considered as the position of each particle.Step 2: Generate MST graph similar to step 2 of BA.Step 3: Set *t* = 0. The following steps are done until t less than the number of generations (*t* < $N_{iteration})$.
Step 3.1: Calculate fitness functions according to Equation (13).Step 3.2: Determine the best particle in the swarm as global best (gbest) that have a higher fitness value and also determine the best position of each particle (pbest). For the first iteration, the initial pbest is considered as the position of the particle.Step 3.3: Update the velocity vector for all particles. The velocity of each particle is defined as 2n × 1 vector. In this vector, odd and even elements represent respectively the x and y velocities of the network vertices. To calculate velocity, first distance matrix D is defined by an n × n′ matrix as follows:
(16)$$D=\left[\begin{array}{ccc} d_{11} & d_{12} & \ldots \\ d_{21} & d_{22} & \ldots \\ \begin{array}{c} \vdots \\ d_{n1} \end{array} & \begin{array}{c} \vdots \\ d_{n2} \end{array} & \begin{array}{c} \ddots \\ \ldots \end{array} \end{array}\begin{array}{c} d_{1n^{\prime }}\\ d_{2n^{\prime }}\\ \begin{array}{c} \vdots \\ d_{nn^{\prime }} \end{array} \end{array}\right]\;\to \;D=\left[\begin{array}{c} \begin{array}{c} \Delta x_{1}\\ \Delta y_{1}\\ \vdots \end{array}\\ \Delta x_{n}\\ \Delta y_{n} \end{array}\right]$$
where {1, 2, …, n} are the sensor node numbers of each particle and {1, 2, …, n′} are the sensor node numbers of the gbest particle (or pbest)). Second, the smallest element is computed in each matrix row, which represents the minimum distance. Third, the *x* and *y* differences between each vertex and selected vertex of pest (or gbest) particles are calculated. The velocity vector is calculated to pbest and gbest particles separately. Finally, the total velocity is calculated by Equation (17).
(17)$$\vec{v_{i}}=w\vec{v_{i}}+c_{1}\;\vec{\varphi_{1i}}\;\vec{d_{\mathrm{pbest}}}+c_{2}\vec{\varphi_{2i}}\;\vec{d_{\mathrm{gbest}}}$$
where *w* is the inertia weight, *c*_1_ and *c*_2_ are acceleration coefficient and constant parameter, $\varphi_{1}$ and $\varphi_{2}$ are randomly parameters which are selected for the particles in each step. If the velocity of each particle is obtained at a distance of more than 200 m, it is multiplied by 0.1. It should be noted the initial velocity is considered zero.Step 3.4: Update the position vector for all particles. The position vector of each particle is updated using its velocity vector as below.
(18)$$\left[\begin{array}{c} \begin{array}{c} x_{1}\\ y_{1}\\ \vdots \end{array}\\ x_{n}\\ y_{n} \end{array}\right]=\left[\begin{array}{c} \begin{array}{c} x_{1}\\ y_{1}\\ \vdots \end{array}\\ x_{n}\\ y_{n} \end{array}\right]+\vec{v_{i}}$$Step 3.5: t = t + 1.Step 4: Find the best global bee (WSN).

## 4. Experimental Results and Discussion

To evaluate the sensor network deployment, which is described in Section 3, the BA and POS algorithms are developed and compared using three parameters in this section. To conduct the numerical experiment, two health and environmental application have been addressed: monitoring water quality and monitoring medical equipment in a hospital. In these scenarios, finding optimal positions of the sensor nodes so that the coverage and lifetime are maximized is so challengeable. Monitoring water pollution or water quality of water resources is the top priority issue for sustainable development these days. WSN is the suitable solution to address the issue. Due to the high cost of placement and battery change of the sensors in the water resource especially for underwater resource, it is necessary to deploy WSN with the minimum number of sensors and maximizing the coverage and lifetime. Another scenario is monitoring a wheelchair in a hospital. As the number of medical equipment such as wheelchairs are limited in a hospital, it is important to monitor this equipment. Hospitals are sensitive areas so that constructions (placing and changing the sensor nodes) are not possible at any time and at any place, due to creating pollution. Hence, to deploy WSN for monitoring medical equipment it is important to find not only the optimal positions of the sensors but also the coverage and lifetime of the network. These scenarios indicate the importance of GIS-based optimization algorithms for WSN deployment.

For these scenarios, the experiment area is simulated with 1000 m × 1000 m region where it is required to deploy a fixed number of sensor nodes. In this experiment study, the same parameters were considered for all sensor nodes are considered as for both algorithms. In addition, the parameters of the objective function (Equation (13)) are set same for both algorithms: α = 0.6; β = 0.4. The value of $C_{R}$ in this numerical study is the area of the simulated region, $10^{6}{\;m}^{2}$.
*n* = 20;$R_{c}$ = 200 m;$E_{elec}$ = 50 nJ/bit;*E* = 5 J;$R_{s}$ = 300 m;$\in_{amp}$ = 100 pJ/bit;

The major step of applying evolutionary algorithms is algorithm design for solving the WSN deployment in health and environmental applications. Parameter calibration, for instance population size and number of iterations to choose right parameter values, is an important kind of algorithm design since the parameter values have an effect on the performance of the algorithms [86,87,88]. One of the methods for the parameter calibration is the parameter tuning in which the parameter values are chosen in the initialization step and don’t change when the algorithm is running [86,88]. Therefore, two algorithms were implemented with different parameter values in the simulated region to calibrate the parameters. The BA parameters that are needed to be set are the number of iteration, employed bees, better bees, elite bees, onlooker bees and neighborhood search. For PSO algorithm, the number of iteration and particles as well as inertia weight, acceleration coefficient and constant parameter are needed to be calibrated. The results of the implementation of BA and PSO algorithms with different values are presented in Table 1 and Table 2, respectively.

Table 1 indicates that by increasing the searching distance of the better bees (dbp), the fitness values were not getting better, but when that of the best bees (nep) was increased until value 10, the fitness values were changed toward a better value. This issue was contrary to the number of onlooker bees. In other words, increasing the number of onlooker bees for elite and better bees (nep and nbp) leads to better fitness values. Finally, row 20, which has better fitness value, was selected as the parameter values of BA. Table 2 shows that increasing the value of *c_1_* and *c_2_* results in the better fitness value, while by increasing the value of the w the fitness value was decreased. As a result, the parameter values of row 20 were selected as best for PSO implementation.

The experimental studies were implemented using Java programming language and GeoTools library (for performing GIS analyses). To evaluate and compare two BA- and PSO-based sensor network deployment algorithms, three criteria including convergence rate, repeatability, and modeling complexity are used. The results of these evaluation criteria are summarized in the following subsections.

### 4.1. Convergence Rate

One of the important criteria for the suitability of a meta-heuristic algorithm is its convergence rate [89]. This criterion is selected to analyze the performance of the both algorithms for the IoT-based network deployment. Figure 6 shows the global fitness values of both algorithms for 20 different executions for the WSN deployment with optimizing the coverage and the lifetime. In the figure, the results of each execution for different iterations are shown with a line. In each iteration of the BA and PSO algorithms, the positions of the sensors were randomly changed (Equations (14) and (15)) with consideration of the connectivity constraint and creating the MST graph between the sensor nodes and then the fitness value was calculated by computing the network coverage and lifetime (Equations (3), (4), and (9)). The global fitness of each iteration is the best fitness value that was obtained within the population size of the algorithms. The global fitness has changed from 0.627 to 0.755 in 100 iterations of BA while it has changed from 0.625 to 0.817 in 70 iterations of PSO algorithm (in the best iteration). Based on the figure, both algorithms have a higher convergence rate in the first iterations and in the subsequent runs. The convergence rate is reduced and continued smoothly until the global fitness is found. However, the convergence rate of the BA faster than that of PSO algorithm in the first ten iterations because of using neighborhood searching around elite and better bees by onlooker bees in BA algorithm. In general, the convergence rate of PSO is smoother than that of BA in all the iterations. The PSO algorithm is converged in the first 30 iterations. The reason for this issue could be small changes in the particles in each iteration. In addition, this issue shows that the PSO algorithm has better performance than BA in dealing with local optimums.

The average of the best fitness values for each iteration obtained by 20 different executions are shown in Figure 7. The figure shows, in general, the convergence of the PSO is better than that of BA and thus is better performance for network deployment for environmental and health sensors. The maximum average of fitness obtained 0.764 using the PSO algorithm and 0.727 using BA while the minimum average values of the two algorithms obtained approximately the same 0.64. In the 20 first iterations, BA found better values and in the other iterations PSO found better values. In other words, the PSO algorithm in the 20 first iterations converges to lower values of the fitness function than BA, while in the other iterations it convergence to much higher values; the best fitness value found by the PSO algorithm in the integration is around 70, but was not found by BA. These results indicate that the PSO algorithm, approximately 5% on average, gained the better positions of the sensors regarding the trade-off between the network coverage and lifetime than BA.

### 4.2. Constancy Repeatability

Another criterion that is much more important than the convergence rate is the stability and repeatability of the algorithm. The similarity rate of the results for different executions (implementations) with the same input values is called algorithm constancy repeatability [89]. In this study, it was calculated by executing both algorithms 20 times. Figure 8 shows the best result of sequential executions of the algorithms. According to the figure, the changes in the best fitness values for the PSO algorithm are greater than those of BA in the 20 executions. These changes are from 0.702 to 0.752 for BA and from 0.713 to 0.817 for the PSO algorithm. To more accurately calculate the constancy, the standard deviation of the results of the sequential executions is obtained as follows:(19)STDEV (f(x))= ∑ii(fibest−μ)2ii
where *ii* is the number of executions, $f_{i}^{best}$ is the best global fitness of each execution, and μ is the mean of the results.

The value of the variance will be obtained between zero and one, as the result values are normal; the values are between zero and one. Whenever the variance is closer to zero, the algorithm is more constant. The standard deviation for both BA and PSO algorithms was obtained 0.0156 and 0.0338, respectively. Although the variance value of the BA is lower than that of PSO, both values are low and near to zero. This proves the constancy and repeatability of both algorithms for the GIS-based network deployment problem.

### 4.3. Modeling Complexity

By comparing the BA and PSO algorithms, it is determined that the number of parameters needed to be set by the user for PSO is fewer than for BA. In PSO, the user is required to determine the values of three parameters before the start of the algorithm, in addition to the run number, population size and the coefficients of fitness function (α, β) which exist in both algorithms. In contrast to that, six parameters are needed for value-initialization before the BA starts (since the number of better bees depends on the number of scout bees, there are six independent parameters in BA). Although both algorithms are easy to implement, the formulation of the problem in BA is slightly more complex than in PSO. The reason is that the same operation is done for all particles of the swarm, while there are three types of bees in BA. In addition, the number of new bees (scout bees) is created in each run in BA. On the other hand, the particles are created only once at the beginning of the algorithm and their positions are updated during the algorithm. As parameter setting is not an easy task, and determining proper values for the parameters by the stakeholders affects the success of the algorithm, an algorithm with lower parameters is preferred [90]. In this way, the PSO algorithm in this study has better performance than BA for IoT-based network deployment.

## 5. Discussion

In this study, GIS-based optimization algorithms have been applied for WSN deployment in health and environmental applications. GIS has several advantages in WSN deployment. GIS with modeling sensor nodes as point geometries makes WSN deployment simple due to simply managing the positions and topology between sensors. Another advantage of GIS is spatial analyses such as proximity analyses that make it possible to calculate the total unique coverage of the WSN (by removing overlapping coverage of the sensors) precisely. Applying the MST routing protocol not only can implement in the GIS easily due to modeling positions of the sensors but also help to find optimal topology of WSN with the minimum length for reducing energy consumption and then prolonging the lifetime. In addition, the evolutionary optimization algorithms, BA and PSO algorithms, help to find optimum positions of the sensors in WSN with trading-off between the coverage and lifetime. Therefore, the GIS-based optimization algorithm is a suitable solution to find the optimal spatial distribution of the sensors with a limited number of sensors in a region.

PSO algorithms help decision makers of health and environment fields to manage the WSN deployment in their scenarios by finding the optimal spatial distribution of the sensors as PSO algorithm has less complexity and better convergence rate than BA as well as has good performance in repeatability. The better performance of PSO is due to the continuity of the problem of WSN deployment. Based on the research [91,92], the PSO algorithm is suitable for continuous optimization problems. WSN deployment is a continuous problem as the sensor nodes can be located at any position in the region. In addition, PSO can find better positions for sensors and the network topology with more trade-off between the network coverage and lifetime.

Finally, the GIS-based optimization algorithm for WSN deployment has advantages in health and environmental application, although it seems simply WS deployment. This research highlights two important conflict issues of WSN deployment, coverage and lifetime, in the health and environmental applications. These issues are modeled using the GIS-based optimization algorithm in this research. In addition, this algorithm helps to enable IoT-based system and smart cities, because it is possible to provide ubiquitous health and environment services at any time and at any place with maximizing coverage and lifetime.

## 6. Conclusions and Future Works

Careful WSN deployment is necessary to build reliable WSN in different environmental and health applications to provide ubiquitous services at any place and at any time. The first step to monitor environmental and health parameters using WSN is to find the locations of the minimum number of the sensor nodes in the target region including hospitals, smart cities, smart homes, forests, and water resources so that sensor nodes cover the region. To cover the region, the health and environmental sensor nodes should be located according to the sensing ranges of the sensor nodes. However, the distance between the sensor nodes cannot be long (especially longer than the communication ranges of the sensor nodes) because the energy consumption of the sensor nodes increases with increasing the distances. As the sensors, such as GPS receiver, humidity, temperature, pressure, and proximity sensors that are utilized in different health and environmental applications have a certain power consummation range, this results in a limited lifetime. The limited lifetime of the sensors makes the network fail to provide ubiquitous services in health and environmental applications in which changing of the batteries of the sensor nodes is challenging and costly (such as underwater resources). Therefore, the coverage and lifetime should be optimized in the WSN deployment of health and environmental applications.

In this research, the WSN deployment in environmental and health applications are reviewed. Environmental applications including water, air pollution, flood, agriculture, earthquake, landslide, habitat, soil, temperature, and fire monitoring are the applications where optimization of WS deployment is important. In the health domain, telemonitoring, smart hospitals to track patients, medical equipment, and medicine, emergency ambulance services, and smart home are the main applications where careful WSN deployment affects the performance of these networks. In addition, this research compared the two optimization algorithms (BA and PSO) for performance analysis of IoT-based sensor network deployment with optimizing the network coverage and lifetime for environmental and health applications. Selecting the proper position for sensor nodes is an important problem in the network deployment to maximize the environment or the number of patients is covered by the network to have effective health and environmental planning. Increasing coverage has effects on the patient and environment security and safety. The experiment of the GIS-based deployment was performed in the simulated region with the fixed number of sensor nodes for testing both algorithms. As MST with a minimum length was used, it results in minimum energy consumption, according to ET.

The results of the GIS-based sensor network deployment in the simulated region showed the applicability of both algorithms. However, the PSO-based network deployment algorithm generated much better results than BA. The convergence rate of the PSO algorithm toward the optimum solution was gradual, but the convergence of BA occurred faster in the first iteration. The PSO algorithm also converges to better fitness values against the BA-based network deployment algorithm. In addition, the results indicate that the repeatability of both algorithms is reasonable, since the obtained values of the standard deviation, 0.0150 and 0.0375, are near zero for both algorithms.

Although this research is efficient in optimizing network deployment using GIS analysis for environmental and health applications, it has some limitations that need to be considered in future work. The number of the sensor nodes is fixed in this research, while it can be optimized as another objective of the algorithm. WSN deployment using heterogeneous health and environment sensor nodes is another future research. Clustered MST algorithm can be evaluated to improve the proposed algorithm as the future study because the sensor network clustering has more benefits such as scalability, avoiding redundant data, latency [93]. Evaluating other algorithms, including genetic and ant colony, for WSN deployment in health and environment applications is another future research.

## Figures and Tables

**Figure 1 sensors-20-05923-f001:**
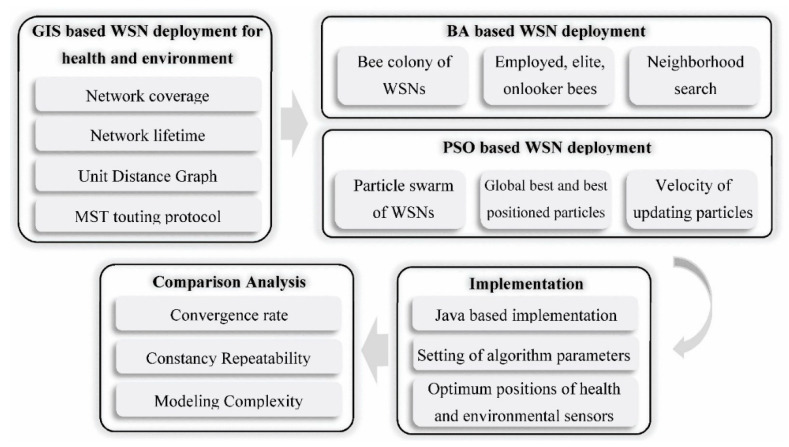
The research methodology.

**Figure 2 sensors-20-05923-f002:**
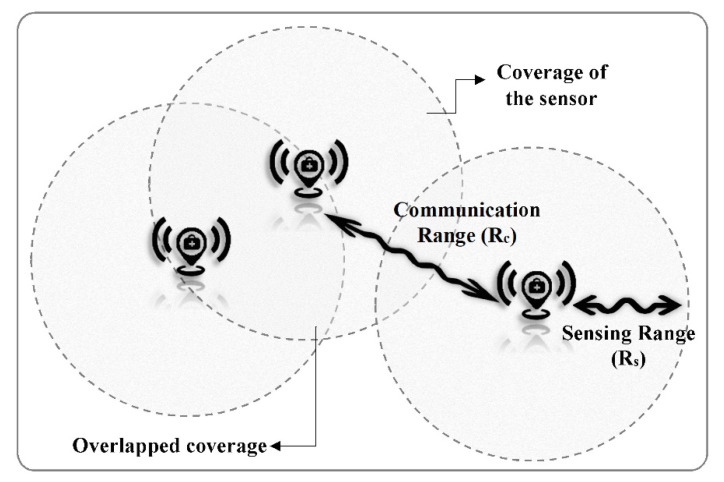
Sensor nodes: Sensing range, communication range, and coverage.

**Figure 3 sensors-20-05923-f003:**
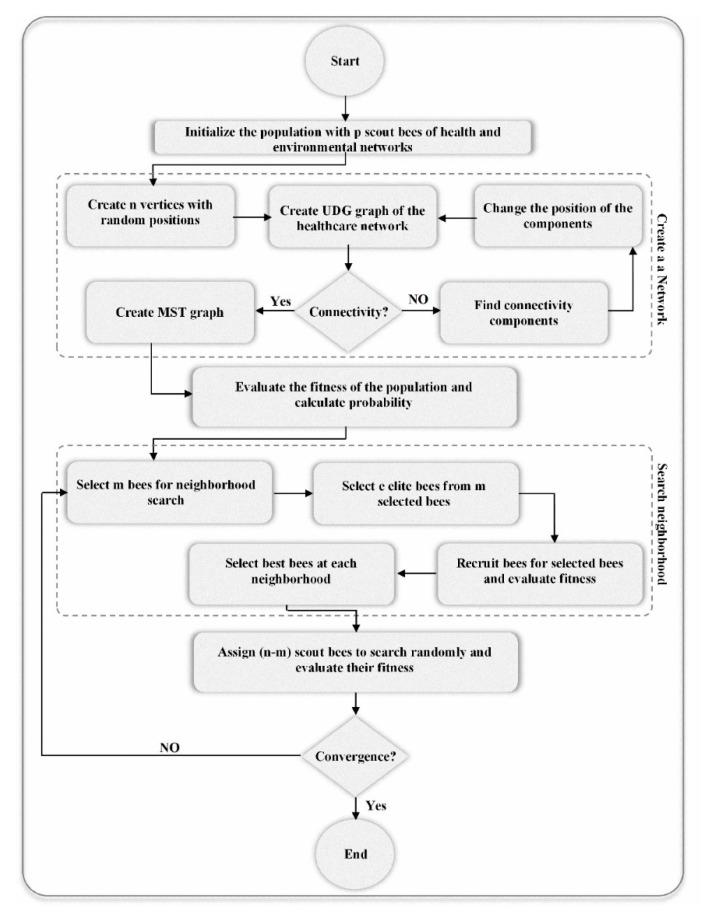
Bees Algorithm (BA) for Minimum Spanning Tree (MST)-based network deployment.

**Figure 4 sensors-20-05923-f004:**
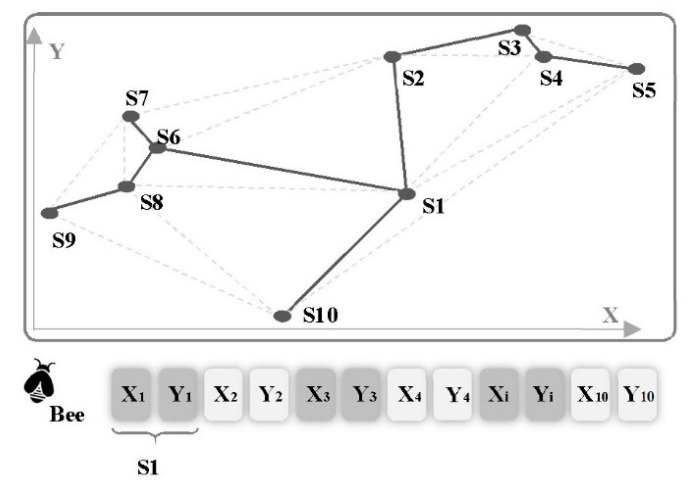
Schematic example of Bee with *n* = 10.

**Figure 5 sensors-20-05923-f005:**
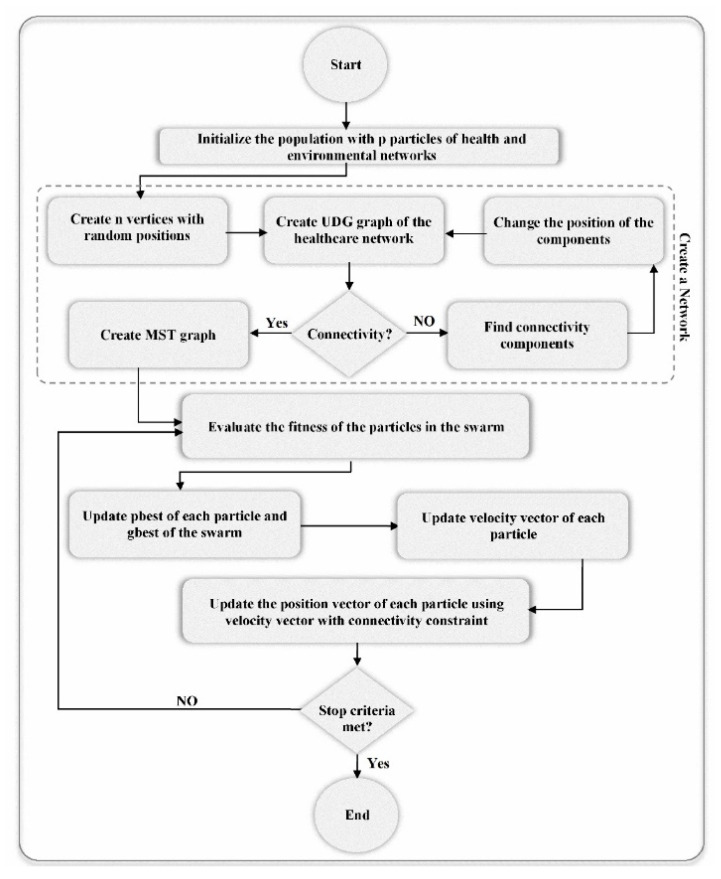
Particle Swarm Optimization (PSO) for MST-based network deployment.

**Figure 6 sensors-20-05923-f006:**
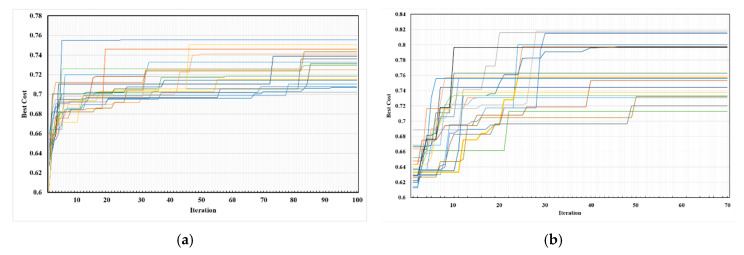
Convergence rate of two network deployment algorithms for 20 executions: (**a**) BA; (**b**) PSO.

**Figure 7 sensors-20-05923-f007:**
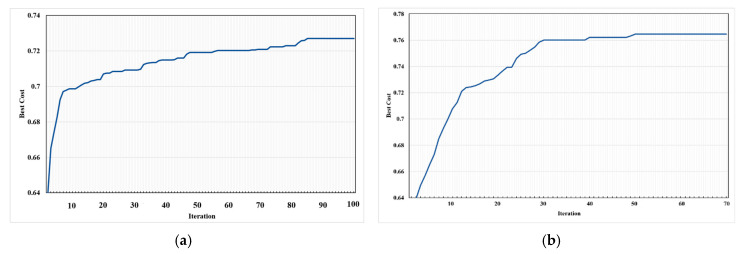
The average of the best values over 20 executions for two algorithms: (**a**) BA; (**b**) PSO.

**Figure 8 sensors-20-05923-f008:**
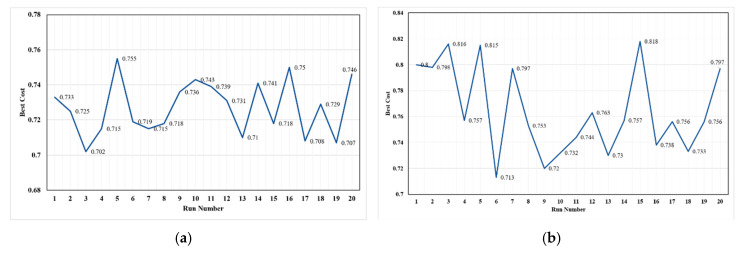
Constancy of two algorithms for IoT-based sensor network deployment algorithm in sequential executions: (**a**) BA; (**b**) PSO.

**Table 1 sensors-20-05923-t001:** Calibration of the BA parameters using trial and error method.

Run No.	Iteration Number	*p*	*m*	*e*	*s*	*nep*	*nbp*	*dep*	*dbp*	Fitness Value
1	20	100	35	10	55	5	5	5	5	0.711
2	20	100	35	10	55	12	5	5	5	0.715
3	20	100	35	10	55	20	5	5	5	0.721
4	20	100	35	10	55	12	12	5	5	0.713
5	20	100	35	10	55	12	20	5	5	0.740
6	20	100	35	10	55	12	20	10	5	0.742
7	20	100	35	10	55	12	20	12	5	0.718
8	20	100	35	10	55	12	20	20	5	0.699
9	20	100	35	10	55	12	20	12	10	0.719
10	20	100	35	10	55	12	20	10	12	0.715
11	20	100	35	10	55	12	20	10	10	0.726
12	20	100	40	5	55	12	20	10	5	0.693
13	20	100	30	15	55	12	20	10	5	0.697
14	20	100	40	10	50	12	20	10	5	0.715
15	20	50	17	5	28	12	20	10	5	0.729
16	20	50	20	5	25	12	20	10	5	0.712
17	20	100	35	10	55	12	20	10	5	0.724
18	50	100	35	10	55	12	20	10	5	0.731
19	70	100	35	10	55	12	20	10	5	0.739
20	100	100	35	10	55	12	20	10	5	0.755 (best)

**Table 2 sensors-20-05923-t002:** Calibration of the PSO parameters using trial and error method.

Run No.	Iteration Number	*p*	*w*	*c_1_*	*c_2_*	Fitness Value
1	20	100	0.3	0.5	0.3	0.636
2	20	100	0.3	1.5	0.3	0.639
3	20	100	0.3	2	0.3	0.657
4	20	100	0.3	4	0.3	0.665
5	20	100	0.3	4	1.5	0.719
6	20	100	0.3	4	2	0.761
7	20	100	0.3	4	4	0.735
8	20	100	0.3	2	2	0.732
9	20	100	0.3	2.5	2	0.703
10	20	100	0.8	2	2	0.683
11	20	100	0.5	2	2	0.681
12	20	100	0.3	2	2	0.761
13	20	50	0.3	2	2	0.720
14	20	20	0.3	2	2	0.665
15	20	120	0.3	2	2	0.799
16	20	80	0.3	2	2	0.763
17	20	150	0.3	2	2	0.719
18	30	100	0.3	2	2	0.767
19	50	100	0.3	2	2	0.797
20	70	100	0.3	2	2	0.816 (best)

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
