# Peer review of "Performance Analysis of IoT-Based Health and Environment WSN Deployment"

_sensors, 2020, doi:10.3390/s20205923_

Round 1

Reviewer 1 Report

The paper deals with comparison of two algorithms used to determine positions of nodes in WSN. The algorithms used in the paper are BA - Bees Algorithm and PSO - Particle Swarm Optimization. 

The English language needs some improvements, for example: 

"where r is the number of receiving, ????? is the transmitter or receiver circuitry..."

This part of the sentence doesn't really make any sense. 

Quality of figures should be improved as text in figures seems to be blurry. 

It is not quite clear what is the novelty of the paper, there have been quite some implementations of optimization algorithms for WSN deployments. Sections that describe the state of the art, should be more focussed on the implementation of optimization algorithms and not on the description of sensors used to measure parameters in other implementations. 

What is the difference between the parameters in the last 3 rows in Table 1? Seems the only difference is the number of iterations. That leads to a question, what would happen if a number of iterations is fixed (let's assume higher number than 100) or if there is some other parameter to stop iterations (i.e. insignificant change in fitness value).

Same applies to Table 2.

Testing different input parameters of the optimization algorithms with the different number of iterations doesn't seem to be the right way to find an optimal value.

Authors mentioned they used GoeTools to perform GIS analysis. However, in the paper, there is no information whether there were any constraints on where sensors can be placed?

Was data from GIS used to modify coverage/communication range of the sensor nodes in WSN?

The performance of the algorithms was evaluated using fitness value, convergence rate and repeatability. However, this doesn't really provide any added value from the WSN performance point of view. 
Were achieved results tested from the network performance point of view? 

Author Response

The Statement:

The paper deals with comparison of two algorithms used to determine positions of nodes in WSN. The algorithms used in the paper are BA - Bees Algorithm and PSO - Particle Swarm Optimization.

Response:       

We appreciate the reviewer for his/her valuable time for reviewing our manuscript and providing us useful comments; they have certainly improved our article.

Reviewer#1, Concern # 1:

The English language needs some improvements, for example:

"where r is the number of receiving, ????? is the transmitter or receiver circuitry..."

This part of the sentence doesn't really make any sense.

Author response & action: Thank you for pointing this out. We submitted the manuscript to an English proofreading service. The mentioned sentence have been corrected as follows.

In the equations, r, , and    represent the number of receiving, the transmitter or receiver circuitry and  the transmit amplifier, respectively.

Reviewer#1, Concern # 2:

Quality of figures should be improved as text in figures seems to be blurry.

Author response & action: We appreciate the reviewer’s comment. We have improved the quality of the figures in the revised version of the manuscript.

Reviewer#1, Concern # 3:

It is not quite clear what is the novelty of the paper, there have been quite some implementations of optimization algorithms for WSN deployments. Sections that describe the state of the art, should be more focussed on the implementation of optimization algorithms and not on the description of sensors used to measure parameters in other implementations.

Author response & action: We appreciate the reviewer’s comment. Based on the reviewer’s comment the following paragraphs have been modified in the revised version of the manuscript.

It reads (abstract):  With the development of Internet of Things (IoT) applications, applying the potential and benefits of IoT technology in the health and environment services is increasing to improve the service quality using sensors and devices. This paper aims to apply GIS based optimization algorithms for optimizing IoT-based network deployment through the use of wireless sensor networks (WSNs) and smart connected sensors for environmental and health applications. First, the WSN deployment research studies in health and environment applications are reviewed including fire monitoring, precise agriculture, telemonitoring, smart home, and hospital. Second, the WSN deployment process is modelled to optimize two conflict objectives, coverage and lifetime, by applying Minimum Spanning Tree (MST) routing protocol with minimum total network lengths. Third, the performance of the Bees Algorithm (BA) and Particle Swarm Optimization (PSO) algorithms are compared for the evaluation of GIS-based WSN deployment in health and environment applications. The algorithms were compared using convergence rate, constancy repeatability, and modeling complexity criteria. The results showed that the PSO algorithm was converged to higher values of objective functions gradually while BA found better fitness values and was faster in the first iterations. The levels of stability and repeatability were obtained high with 0.0150 of standard deviation for PSO and 0.0375 for BA. The PSO also had lower complexity than BA. Therefore, the PSO algorithm obtained better performance for IoT-based sensor network deployment.

It reads (Page 3, Paragraph 3):  This paper aims to apply GIS-based optimization algorithms for WSN deployment in the health and environmental applications for maximizing the coverage and lifetime by considering the connectivity constraint. To achieve this main goal, three sections of research are as follows. First, previous research studies on WSN deployment are reviewed in environmental and health applications. Second, the MST algorithm has been applied in the WSN deployment process to minimize energy consumption. MST algorithm is a routing precool that minimizes WSN energy consumption with minimum total network lengths. Third, two optimization algorithms, BA and PSO algorithms are compared for evaluating the GIS based WSN deployment in health and environmental applications. BA and PSO are the evolutionary optimization algorithms that have been successfully applied for solving different multi-objective problems. To compare the algorithms, three parameters are used: convergence rate, constancy repeatability, and modeling complexity.

Reviewer#1, Concern # 4:

What is the difference between the parameters in the last 3 rows in Table 1? Seems the only difference is the number of iterations. That leads to a question, what would happen if a number of iterations is fixed (let's assume higher number than 100) or if there is some other parameter to stop iterations (i.e. insignificant change in fitness value).

Same applies to Table 2.

Author response & action: We appreciate the reviewer’s comment. Table 1 and Table 2 show only the results of calibration executions to set proper values of the algorithm parameters. For evaluating the two algorithms (in the following subsections: “Convergence Rate”, “Constancy Repeatability”, “Modeling Complexity”), the number of iteration as well as other parameters of the algorithms are fixed in different executions, as the reviewers mentioned. Table 1 shows with increasing iteration number from 20 to 100 the fitness values were getting better. By more than 100 iterations, the fitness values weren’t changed and the algorithms were converged. Therefore, the last row of Table 1 and Table 2 shows the best values for the parameter of each algorithm to find an optimal solution (e.g. locations of health and environmental sensor nodes so that network coverage and lifetime are maximized). The following information has been added to the revised version of the manuscript.

It reads (Page 14, Paragraph 3): The major step of applying evolutionary algorithms is algorithm design for solving the WSN deployment in health and environmental applications. Parameter calibration, for instance population size and number of iterations to choose right parameter values, is an important kind of algorithm design since the parameter values have an effect on the performance of the algorithms [89, 90, 91]. One of the methods for the parameter calibration is the parameter tuning in which the parameter values are chosen in the initialization step and don’t change when the algorithm is running [89, 91]. Therefore, two algorithms were implemented with different parameter values in the simulated region to calibrate the parameters. The BA parameters that are needed to be set are the number of iteration, employed bees, better bees, elite bees, onlooker bees and neighbourhood search. For PSO algorithm, the number of iteration and particles as well as inertia weight, acceleration coefficient and constant parameter are needed to be calibrated. The results of the implementation of BA and PSO algorithms with different values are presented in Table 1 and Table 2, respectively.

It should be noted that as fitness value is the weighted sum of normalized value of the network coverage and lifetime between 0 and 1, small changes in the fitness value can be significant.  

[89] Eiben, A. E., Michalewicz, Z., Schoenauer, M., & Smith, J. E. (2007). Parameter control in evolutionary algorithms. In Parameter setting in evolutionary algorithms (pp. 19-46). Springer, Berlin, Heidelberg.

[90] Masoumi, Z., Van Genderen, J., & Sadeghi Niaraki, A. (2019). An improved ant colony optimization-based algorithm for user-centric multi-objective path planning for ubiquitous environments. Geocarto International, 1-18.

[91] Eiben, A. E., & Smit, S. K. (2011). Parameter tuning for configuring and analyzing evolutionary algorithms. Swarm and Evolutionary Computation, 1(1), 19-31.

Reviewer#1, Concern # 5:

Testing different input parameters of the optimization algorithms with the different number of iterations doesn't seem to be the right way to find an optimal value.

Author response & action: The authors completely agree with the review’s comment. However, the input values of the algorithms are the same and parameter values of the algorithms are different. The input values of the algorithms are e.g. n = 20, E = 5J, = 200 m, = 300 m, = , = , α = 0.6; β = 0.4, and  that were considered the same for both algorithms. As we stated in Concern #4, Table 1 and Table 2 show the results of the parameter calibration to choose proper values for the algorithm parameters. As the parameters of the algorithms are different (like other AI algorithms such as machine learning algorithms), we couldn’t use the same values for the parameters of both algorithms. Therefore, based on different research [89, 90, 91], the parameter values of each algorithm are specified for solving the given problem. One of the parameters of the evolutionary algorithms that is needed to be calibrated is the iteration number, such as Masoumi et al. (2019), Fang et al. (2017), Saeidian et al. (2016).

Eiben, A. E., Michalewicz, Z., Schoenauer, M., & Smith, J. E. (2007). Parameter control in evolutionary algorithms. In Parameter setting in evolutionary algorithms (pp. 19-46). Springer, Berlin, Heidelberg.

Masoumi, Z., Van Genderen, J., & Sadeghi Niaraki, A. (2019). An improved ant colony optimization-based algorithm for user-centric multi-objective path planning for ubiquitous environments. Geocarto International, 1-18.

Eiben, A. E., & Smit, S. K. (2011). Parameter tuning for configuring and analyzing evolutionary algorithms. Swarm and Evolutionary Computation, 1(1), 19-31.

Fang, Z., Li, L., Li, B., Zhu, J., Li, Q., & Xiong, S. (2017). An artificial bee colony-based multi-objective route planning algorithm for use in pedestrian navigation at night. International Journal of Geographical Information Science, 31(10), 2020-2044.

Saeidian, B., Mesgari, M. S., & Ghodousi, M. (2016). Evaluation and comparison of Genetic Algorithm and Bees Algorithm for location–allocation of earthquake relief centers. International Journal of Disaster Risk Reduction, 15, 94-107.

Reviewer#1, Concern # 6:

Authors mentioned they used GoeTools to perform GIS analysis. However, in the paper, there is no information whether there were any constraints on where sensors can be placed?

Was data from GIS used to modify coverage/communication range of the sensor nodes in WSN?

Author response & action: We appreciate the reviewer’s comment. The following information have been added to the revised version of the manuscript.

It reads (Page 6, paragraph 1): As the previous section shows, WSN deployment is one of the important problem in several environmental and health applications with optimizing conflict objectives (coverage and power consumption or lifetime). The aim of WSN deployment is to find the optimal locations of the sensors so that the network coverage and lifetime are maximized. In WSN deployment, GIS plays key role in modeling the sensor that are located in a specific location (x, y) using point geometry and managing the network of the sensors. In addition, spatial analysis of GIS helps to calculate total coverage of the sensors as well as the overlapped coverage in the given region. The locations of the sensors have great impact on the WSN coverage and energy consumption and even cost. If the locations of the sensors is such that they have the maximum distance from each other according to the sensing range (the points have uniform spatial distribution), the network coverage is maximum and the number of sensors is minimum for the WSN deployment. However, increasing the distance between the sensors increases energy consumption. To reduce energy consumption, it is necessary to keep the distance between the sensors as short as possible, which this issue increases the number of sensors and thus the cost. Therefore, the main issue of the WSN deployment is to find the optimal locations of the sensors.

It reads (Page 8, Paragraph 1): By assuming that the existing sensors can rotate 0–360° horizontally [66], each sensor coverage and total network coverage are calculated by spatial analyses. To this end, first, a point geometry layer is created using the locations of the sensors in which a point represents a sensor node. Then, the buffer polygon around the points are created to display the coverage area of the sensor in the sensing distance using buffer analysis. After that, the overlapped area between two polygons are removed from using Intersection and Union analysis. Buffer, Intersection, and Union are the spatial analyses that are performed on different types of the geometry in GIS [87].

Kumar, D., Singh, R. B., & Kaur, R. (2019). Spatial Data Analysis. In Spatial Information Technology for Sustainable Development Goals (pp. 101-113). Springer, Cham.

Reviewer#1, Concern # 7:

The performance of the algorithms was evaluated using fitness value, convergence rate and repeatability. However, this doesn't really provide any added value from the WSN performance point of view.

Were achieved results tested from the network performance point of view?

Author response & action: We appreciate the reviewer’s comment. In this research, the performance of the network was evaluated in optimizing the network coverage and lifetime using the GIS based optimization algorithm. We have compared two optimization algorithms to optimum trade-off between the two conflict objectives of the network. The performance of the network was calculated using the cost function so that the higher value of the cost function, the better the sensor positions, coverage and lifetime for the network. To make it clear, the discussion section have been added to the revised version of the manuscript. It reads (page 18 and 19):

5- Discussion

In this study, GIS based optimization algorithms have been applied for WSN deployment in health and environmental applications. GIS has several advantages in WSN deployment. GIS with modelling sensor nodes as a point geometries makes WSN deployment simple due to simply managing the positions and topology between sensors. Another advantage of GIS is spatial analyses such as proximity analyses that make it possible to calculate the total unique coverage of the WSN (by removing overlapping coverage of the sensors) precisely. Applying the MST routing protocol not only can implement in the GIS easily due to modelling positions of the sensors but also help to find optimal topology of WSN with the minimum length for reducing energy consumption and then prolonging the lifetime. In addition, the evolutionary optimization algorithms, BA and PSO algorithms, helps to find optimum positions of the sensors in WSN with trading-off between the coverage and lifetime. Therefore, the GIS based optimization algorithm is a suitable solution to find the optimal spatial distribution of the sensors with a limited number of sensors in a region.

PSO algorithms help decision makers of health and environment fields to manage the WSN deployment in their scenarios by finding the optimal spatial distribution of the sensors as PSO algorithm has less complexity and better convergence rate than BA as well as has good performance in repeatability. The better performance of PSO is due to the continuity of the problem of WSN deployment. Based on the research [92, 93], the PSO algorithm is suitable for continuous optimization problems. WSN deployment is a continuous problem as the sensor nodes can be located at any position in the region. In addition, PSO can find better positions for sensors and the network topology with more trade-off between the network coverage and lifetime.

Finally, the GIS based optimization algorithm for WSN deployment has advantages in health and environmental application, although it seems simply WS deployment. This research highlights two important conflict issues of WSN deployment, coverage and lifetime, in the health and environmental applications. These issues are modelled using the GIS based optimization algorithm in this research. In addition, this algorithm helps to enable IoT based system and smart cities, because it is possible to provide ubiquitous health and environment services at any time and at any place with maximizing coverage and lifetime.

[92] Luo, G., Zhao, H., & Song, C. (2008, November). Convergence analysis of a dynamic discrete PSO algorithm. In 2008 First International Conference on Intelligent Networks and Intelligent Systems (pp. 89-92). IEEE.

[93] AL-Samarrie, A. K., Alyasiri, H., & AL-Nakkash, A. H. (2016). Proposed Multi-Stage PSO Scheme for LTE Network Planning and Operation. International Journal of Applied Engineering Research, 11(20), 10199-10210.

Reviewer 2 Report

In this paper the authors analyze two multi-objective optimization methods for the IoT-based network deployment through the use of wireless sensor networks (WSNs) and smart connected sensors for environmental and health applications. I do not understand why is a health and environmental sensor network deployment, it is simply a sensor network deployment.

In my opinion the paper must be change the following:

The abstract must be change in its entirety because of it do not is a summary of the paper. In my opinion, an abstract should convey the main results and conclusions of a scientific paper because it communicates complex investigations and can act as an independent entity instead of an entire article. In the case of this article, clearly, this does not happen. It is the very extensive, there are many sentences left over. There are superfluous phrases that must be removed. For instance: “ However, IoT based….studies” or “The two algorithms are… analysis” or “Both algorithms…these criteria”.

Regarding to the Introduction, it does not put in perspective the motivation and objectives.

What is the purpose of the paper? What motivates the authors to write this paper?

These two questions should be clearly answered in the first section of the paper.

In this part, it is important to reference paper: “A neural network model to develop actions in urban complex systems represented by 2D meshes” of International Journal of Computer Mathematics.

In short, after reading the introduction it is not clear what the authors want to do, what the objective is, etc. The motivation is missing, why is the article written? What does it mean that they don't have other articles already written? What are the objective of the paper?

In my opinion the inane of the section 2 should be change to “Literature review”.

Section 2 is excessively long, it is not necessary to describe in detail the work carried out by different authors. It should be a literature review to see what has been done about it.

Figure 1 and 2 do not contribute anything and must be removed. It is NOT a book, it is a scientific article.

There is no methodology but it is not surprising I bet you do not know for sure what is intended with the article. It should be mandatory a methodology section with a flowchart of the process.

Formula 1 must be removed, it does not contribute anything.

Line 230: “tow sensors”, it will be two sensors.

what does formula 2 mean?

In any mathematical formula in which sums appear (formula 3) , the limits must be set.

The formula 3 can be written in matrix form for a more compact view of it.

The coverage is essential in any sensor network not only in health and environment applications and the first sentence of section 3.1.1 is redundant and should be deleted.

Formulas 5, 6 and 7 do not make sense to them, they are simple formulas for the elementary calculus of areas using double integrals. I repeat that it is not a book, it is an article.

In formula 4 the authors use C_R, but in numerical results, it is not used? or is it deliberately omitted? or is it deliberately omitted?

Why do you use these two algorithms (BE and PSO)?

There are no others?

Maybe some neural network algorithm would be better?

Line 319, if^2?

Only one experiment is performed and the results are not discussed.

Figures 7, 8 and 9 are not well explained.

Both algorithms run 10 times, is it enough to do a proper statistical analysis?

The authors say they perform a GIS analysis, where?,

do they know what a GIS analysis is?

The theoretical part of the article has potential, but it must be presented much better, abstract, introduction, methodology and experimental results must be redone or greatly improved.

Other menor review is that throughout the paper, the authors have to avoid personal opinions and non-formal phrases, such as “we believe that”, “for example”, “the data needed”, etc.

It would be necessary to completely redo the paper to present it properly.

Author Response

The Statement:

In this paper the authors analyze two multi-objective optimization methods for the IoT-based network deployment through the use of wireless sensor networks (WSNs) and smart connected sensors for environmental and health applications.

Response:

We appreciate the reviewer for his/her valuable time for reviewing our manuscript and providing us useful comments; they have certainly improved our article.

Reviewer#2, Concern # 1:

I do not understand why is a health and environmental sensor network deployment, it is simply a sensor network deployment.

Author response & action: We appreciate the reviewer’s comment. To make it clear, we have added the research problem in the “Introduction” section and explained health and environmental scenarios in the numerical experiment.

It reads (Page 1, Paragraph 1): In environmental and health monitoring applications, the WSNs have to be designed extremely reliably because it can be literally the difference between life or death. It means that they have to be designed in such a way that they ubiquitously provide the environmental and health services at any place and at any time. To deploy such a WSN, due to the high cost of the sensors, the minimum number of sensors has to be located in the position to create maximum coverage of the region. On the other hand, the batteries of the sensors limit the provision of the health and environmental services ubiquitously. This issue creates more challenges for WSN deployment in the environment and health applications due to the difficult access areas in the environment (e.g. mountains, forests, and groundwater) as well as the sensitivity areas in the hospitals where it is not possible to replace the sensors' batteries continuously. To consider this issue, WSN has to be deployed so that the sensors consume less energy. Therefore, coverage and energy consumption are two important issues in the WSN deployment in environmental and health applications.

It reads (Page 13, Paragraph 5): To conduct the numerical experiment, two health and environmental application have been addressed: monitoring water quality and monitoring medical equipment in a hospital. In these scenarios, finding optimal positions of the sensor nodes so that the coverage and lifetime are maximized is so challengeable. Monitoring water pollution or water quality of water resources is the top priority issue for sustainable development these days. WSN is the suitable solution to address the issue. Due to the high cost of placement and battery change of the sensors in the water resource especially for underwater resource, it is necessary to deploy WSN with the minimum number of sensors and maximizing the coverage and lifetime. Another scenario is monitoring a wheelchair in a hospital. As the number of medical equipment such as wheelchairs are limited in a hospital, it is important to monitor this equipment. Hospitals are sensitive areas so that constructions (placing and changing the sensor nodes) are not possible at any time and at any place, due to creating pollution. Hence, to deploy WSN for monitoring medical equipment it is important to find not only the optimal positions of the sensors but also the coverage and lifetime of the network. These scenarios indicate the importance of GIS based optimization algorithms for WSN deployment.

In my opinion the paper must be change the following:

Reviewer#2, Concern # 2:

The abstract must be change in its entirety because of it do not is a summary of the paper. In my opinion, an abstract should convey the main results and conclusions of a scientific paper because it communicates complex investigations and can act as an independent entity instead of an entire article. In the case of this article, clearly, this does not happen. It is the very extensive, there are many sentences left over. There are superfluous phrases that must be removed. For instance: “ However, IoT based….studies” or “The two algorithms are… analysis” or “Both algorithms…these criteria”.

Author response & action: We appreciate the reviewer’s comment. The abstract have been modified in the revised version of the manuscript.

With the development of Internet of Things (IoT) applications, applying the potential and benefits of IoT technology in the health and environment services is increasing to improve the service quality using sensors and devices. This paper aims to apply GIS based optimization algorithms for optimizing IoT-based network deployment through the use of wireless sensor networks (WSNs) and smart connected sensors for environmental and health applications. First, the WSN deployment research studies in health and environment applications are reviewed including fire monitoring, precise agriculture, telemonitoring, smart home, and hospital. Second, the WSN deployment process is modelled to optimize two conflict objectives, coverage and lifetime, by applying Minimum Spanning Tree (MST) routing protocol with minimum total network lengths. Third, the performance of the Bees Algorithm (BA) and Particle Swarm Optimization (PSO) algorithms are compared for the evaluation of GIS-based WSN deployment in health and environment applications. The algorithms were compared using convergence rate, constancy repeatability, and modeling complexity criteria. The results showed that the PSO algorithm was converged to higher values of objective functions gradually while BA found better fitness values and was faster in the first iterations. The levels of stability and repeatability were obtained high with 0.0150 of standard deviation for PSO and 0.0375 for BA. The PSO also had lower complexity than BA. Therefore, the PSO algorithm obtained better performance for IoT-based sensor network deployment.

Reviewer#2, Concern # 3:

Regarding to the Introduction, it does not put in perspective the motivation and objectives.

What is the purpose of the paper? What motivates the authors to write this paper?

These two questions should be clearly answered in the first section of the paper.

Author response & action: We appreciate the reviewer’s comment. The following information have been added to the revised version of the manuscript to answer the reviewer’s questions.

It reads (Page 1, Paragraph 1): In environmental and health monitoring applications, the WSNs have to be designed extremely reliably because it can be literally the difference between life or death. It means that they have to be designed in such a way that they ubiquitously provide the environmental and health services at any place and at any time. To deploy such a WSN, due to the high cost of the sensors, the minimum number of sensors has to be located in the position to create maximum coverage of the region. On the other hand, the batteries of the sensors limit the provision of the health and environmental services ubiquitously. This issue creates more challenges for WSN deployment in the environment and health applications due to the difficult access areas in the environment (e.g. mountains, forests, and groundwater) as well as the sensitivity areas in the hospitals where it is not possible to replace the sensors' batteries continuously. To consider this issue, WSN has to be deployed so that the sensors consume less energy. Therefore, coverage and energy consumption are two important issues in the WSN deployment in environmental and health applications.

It reads (Page 2, Paragraph 3):  Although such research has addressed both important issues in the sensor network deployment, these studies haven’t compared two evolutionary optimization algorithms (Bees Algorithm (BA) and Particle Swarm Optimization (PSO) algorithms) for WSN deployment with considering coverage, lifetime and connectivity objectives. In addition, they haven’t used a Minimum Spanning Tree (MST) algorithm for path and energy optimization in WSN deployment.

It reads (Page 3, Paragraph 3):  This paper aims to apply GIS-based optimization algorithms for WSN deployment in the health and environmental applications for maximizing the coverage and lifetime by considering the connectivity constraint. To achieve this main goal, three sections of research are as follows. First, previous research studies on WSN deployment are reviewed in environmental and health applications. Second, the MST algorithm has been applied in the WSN deployment process to minimize energy consumption. MST algorithm is a routing precool that minimizes WSN energy consumption with minimum total network lengths. Third, two optimization algorithms, BA and PSO algorithms are compared for evaluating the GIS based WSN deployment in health and environmental applications. BA and PSO are the evolutionary optimization algorithms that have been successfully applied for solving different multi-objective problems. To compare the algorithms, three parameters are used: convergence rate, constancy repeatability, and modeling complexity.

It reads (Page 4, Paragraph 1):  This article is organized as follows. Section 2 reviews WSN deployment researches in several applications of environmental and health applications and shows the importance of coverage and lifetime in WSN deployment for these fields. Section 3 describes the methodology of the research.  This section includes the GIS based WSN deployment method to optimize the network coverage and lifetime with applying MST routing algorithm. In addition, the steps of BA and PSO optimization algorithms for GIS based WSN deployment are explained in this section. Section 4 covers how the algorithms implemented in a simulated health and environmental region and experimental results of comparing two algorithms using three criteria. Section 5 belongs to the discussion of the research. Finally, conclusions and future works are explained in Section 6. 

Reviewer#2, Concern # 4:

In this part, it is important to reference paper: “A neural network model to develop actions in urban complex systems represented by 2D meshes” of International Journal of Computer Mathematics.

Author response & action: Based on the reviewer’s suggestion, the following information have been added to the revised version of the manuscript. It reads (Page 1, Paragraph 1):

It has to be highlighted that the Wireless Sensor Network (WSN) plays a significant role in IoT because it can be developed in many fields [5, 6], so that the most cities have deployed or are planning to deploy WSN for creating city-wide wireless coverage [86].

[86] Oliver, J. L., Tortosa, L., & Vicent, J. F. (2011). A neural network model to develop actions in urban complex systems represented by 2d meshes. International Journal of Computer Mathematics, 88(16), 3361-3379.

Reviewer#2, Concern # 5:

In short, after reading the introduction it is not clear what the authors want to do, what the objective is, etc. The motivation is missing, why is the article written? What does it mean that they don't have other articles already written? What are the objective of the paper?

Author response & action: As we answered for Concern #3, the objective paragraph of the Introduction section have been modified in the revised version of the manuscript.

It reads (Page 3, Paragraph 3):  This paper aims to apply GIS-based optimization algorithms for WSN deployment in the health and environmental applications for maximizing the coverage and lifetime by considering the connectivity constraint. To achieve this main goal, three sections of research are as follows. First, previous research studies on WSN deployment are reviewed in environmental and health applications. Second, the MST algorithm has been applied in the WSN deployment process to minimize energy consumption. MST algorithm is a routing precool that minimizes WSN energy consumption with minimum total network lengths. Third, two optimization algorithms, BA and PSO algorithms are compared for evaluating the GIS based WSN deployment in health and environmental applications. BA and PSO are the evolutionary optimization algorithms that have been successfully applied for solving different multi-objective problems. To compare the algorithms, three parameters are used: convergence rate, constancy repeatability, and modeling complexity.

Reviewer#2, Concern # 6:

In my opinion the inane of the section 2 should be change to “Literature review”.

Section 2 is excessively long, it is not necessary to describe in detail the work carried out by different authors. It should be a literature review to see what has been done about it.

Author response & action: We appreciate the reviewer’s comment. We have changed the title of the section 2 to “Literature review” and we have removed the unnecessary sentences in this section.

Reviewer#2, Concern # 7:

Figure 1 and 2 do not contribute anything and must be removed. It is NOT a book, it is a scientific article.

Author response & action: Figure 1 and 2 have been removed from the revised version of the manuscript.

Reviewer#2, Concern # 8:

There is no methodology but it is not surprising I bet you do not know for sure what is intended with the article. It should be mandatory a methodology section with a flowchart of the process.

Author response & action: Based on the reviewer’s comment, the following modifications have been applied in the revised version of the manuscript. It reads (Page 5, paragraph 2):

Based on what can be inferred from the previous research in several applications of health and environment (section 2), the sensor network deployment is a multi-objective optimization problem to maximize coverage and lifetime with considering the connectivity constraint. To solve this multi-objective optimization problem, GIS based optimization algorithms are applied in this research. The methodology of applying and evaluating GIS based optimization algorithms to optimize coverage and lifetime in health and environmental applications are shown in Figure 1. In the first step, GIS based optimization is modeled for the WSN deployment. In this regard, coverage and lifetime functions are presented to calculate the total coverage and lifetime of the network. The MST routing protocol is applied to generate a unit distance graph of the sensors to optimize the topology and then minimize the energy consumption. Second, the BA optimization algorithm is designed to step by step deploy health and environmental sensors. BA is one of the artificial algorithms that has been most extensively used in many real-world applications [79] and it is an efficient algorithm that needs fewer control parameters and easy to implement [72, 80]. Third, the PSO algorithm procedure is designed to solve GIS based multi-objective WSN deployment. PSO is one of the swarm intelligence algorithms, which simulates the behavior of bird flocks and fish schools [81-82]. Fourth, the BA and PSO algorithms are implemented for a simulated region. Before testing the algorithms, the algorithm parameters are set using the parameter tuning method. Finally, two BA and PSO algorithms are compared with three criteria.

Figure 1. The research methodology.

 We have changed the title of “Problem definition” subsection to “GIS based WSN deployment”.

Reviewer#2, Concern # 9:

Formula 1 must be removed, it does not contribute anything.

Author response & action: We appreciate the reviewer’s comment. Formula 1 have been removed from the revised version of the manuscript.

Reviewer#2, Concern #10:

Line 230: “tow sensors”, it will be two sensors.

Author response & action: Thank you for pointing out this typos. We submitted the manuscript to an English proofreading for an additional proofread. The probable editorial mistakes have been corrected in the revised version of the manuscript.

Reviewer#2, Concern #11:

what does formula 2 mean?                              

Author response & action: We appreciate the reviewer’s comment. The paragraph have been modified in the revised version of the manuscript according to Formula 1. It reads (Page 7, paragraph 2):

Each edge has m attributes denoted by to represent distance, transmitting energy and so on. In addition, the connectivity between two nodes (node i and j) is defined with . The value of  is one if there is an edge ( ) between the nodes and the edge is selected, otherwise zero.

Reviewer#2, Concern #12:

In any mathematical formula in which sums appear (formula 3) , the limits must be set.

The formula 3 can be written in matrix form for a more compact view of it.

Author response & action: We appreciate the reviewer’s comment. Formula 3 have been modified in the revised version of the manuscript. It reads (Page 7).

Reviewer#2, Concern #13:

The coverage is essential in any sensor network not only in health and environment applications and the first sentence of section 3.1.1 is redundant and should be deleted.

Author response & action: We appreciate the reviewer’s comment. The mentioned sentence have been removed from the revised version of the manuscript.

Reviewer#2, Concern #14:

Formulas 5, 6 and 7 do not make sense to them, they are simple formulas for the elementary calculus of areas using double integrals. I repeat that it is not a book, it is an article.

Author response & action: We appreciate the reviewer’s comment. Based on the reviewer’s comment, the Formula 5, 6,7 have been removed from the revised version of the manuscript.

Reviewer#2, Concern #15:

In formula 4 the authors use C_R, but in numerical results, it is not used? or is it deliberately omitted? or is it deliberately omitted?

Author response & action: As two objectives (coverage and lifetime) have been optimized in our research using Formula 3, it was needed to normalize the values of these objectives. In our research, we have normalized the values of coverage and lifetime between 0 and 1. The CR that is the total area of the region of interest has been used to normalize the total coverage. Therefore, as the obtained results for the best fitness values are in range 0 and 1 (Fig. 7), the normalization has been used in the numerical results. The following information has been added to the revised version of the manuscript.

It reads (Page 7, paragraph 5): where is the coverage of the ith sensor, is the overlapped coverage between the ith sensor and other sensors, is the total area of the region of interest (ROI). In this formula, CR is used to normalize the total coverage. So, the total coverage is a normalized value between 0 and 1.

It reads (Page 14, paragraph 2): In addition, the parameters of the objective function (Eq. 13) are set same for both algorithms: α = 0.6; β = 0.4. The value of CR in this numerical study is the area of the simulated region, .

Reviewer#2, Concern #16:

Why do you use these two algorithms (BE and PSO)?

There are no others?

Maybe some neural network algorithm would be better?

Author response & action: We appreciate the reviewer’s comment. In our research, WSN deployment is defined as a multi-objective optimization problem with maximizing two conflict objectives: coverage and lifetime. Based on Sindhya et al. (2012), multi-criteria decision making (MCDM) and evolutionary multi-objective optimization (EMO) algorithms are applied to solve multi-objective optimization problems. In this research, we focus on EMO algorithms that have widely utilized for solving multi-objective optimization problems in different fields. There are several evolutionary optimization algorithms including Genetic, Ant Colony, PSO, Bee Colony, tabu search (Simon, 2013). As the performances of EMO algorithms are dependent on the problem, in this research, we have compared the performance of two evolutionary algorithms: BA and PSO. As the neural network is not evolutionary optimization algorithms and we haven’t had the sample data to train the neural network, this algorithm hasn’t used in our research. In addition, based on Rafiq et al. (2001), the neural network algorithms are usually applied for classification, filtering, identification, control, and prediction problems. Evaluating other evolutionary algorithms have been added to the revised version of the manuscript.

It reads (Page 2, paragraph 1): Based on [88], multi-criteria decision making (MCDM) and evolutionary multi-objective optimization (EMO) algorithms are applied to solve multi-objective optimization problems. This research focuses on EMO algorithms that have widely utilized for solving multi-objective optimization problems in different fields.

It reads (Page 20, paragraph 2):  Clustered MST algorithm can be evaluated to improve the proposed algorithm as the future study because the sensor network clustering has more benefits such as scalability, avoiding redundant data, latency [85]. Evaluating other algorithms including genetic and an ant colony for WSN deployment in health and environment applications is another future research.

[88] Sindhya, K., Miettinen, K., & Deb, K. (2012). A hybrid framework for evolutionary multi-objective optimization. IEEE Transactions on Evolutionary Computation, 17(4), 495-511.

Simon, D. (2013). Evolutionary optimization algorithms. John Wiley & Sons.

Abraham, A., & Jain, L. (2005). Evolutionary multiobjective optimization. In Evolutionary Multiobjective Optimization (pp. 1-6). Springer, London.

Rafiq, M. Y., Bugmann, G., & Easterbrook, D. J. (2001). Neural network design for engineering applications. Computers & Structures, 79(17), 1541-1552.

Reviewer#2, Concern #17:

Line 319, if^2?

Author response & action: Thank you for pointing this out. The sentence have been modified in the revised version of the manuscript. It reads (Page 9, paragraph 4):

The connectivity of the generated random graph is checked by using its adjacency matrix (G) if and only if [64].

Reviewer#2, Concern #18:

Only one experiment is performed and the results are not discussed.

Author response & action: We appreciate the reviewer’s comment. The discussion section have been added to the revised version of the manuscript. It reads (page 18 and 19):

5- Discussion

In this study, GIS based optimization algorithms have been applied for WSN deployment in health and environmental applications. GIS has several advantages in WSN deployment. GIS with modelling sensor nodes as point geometries makes WSN deployment simple due to simply managing the positions and topology between sensors. Another advantage of GIS is spatial analyses such as proximity analyses that make it possible to calculate the total unique coverage of the WSN (by removing overlapping coverage of the sensors) precisely. Applying the MST routing protocol not only can implement in the GIS easily due to modelling positions of the sensors but also help to find optimal topology of WSN with the minimum length for reducing energy consumption and then prolonging the lifetime. In addition, the evolutionary optimization algorithms, BA and PSO algorithms, helps to find optimum positions of the sensors in WSN with trading-off between the coverage and lifetime. Therefore, the GIS based optimization algorithm is a suitable solution to find the optimal spatial distribution of the sensors with a limited number of sensors in a region.

PSO algorithms help decision makers of health and environment fields to manage the WSN deployment in their scenarios by finding the optimal spatial distribution of the sensors as the PSO algorithm has less complexity and better convergence rate than BA as well as has good performance in repeatability. The better performance of PSO is due to the continuity of the problem of WSN deployment. Based on the research [92, 93], the PSO algorithm is suitable for continuous optimization problems. WSN deployment is a continuous problem as the sensor nodes can be located at any position in the region. In addition, PSO can find better positions for sensors and the network topology with more trade-off between the network coverage and lifetime.

Finally, the GIS based optimization algorithm for WSN deployment has advantages in health and environmental application, although it seems simply WS deployment. This research highlights two important conflict issues of WSN deployment, coverage and lifetime, in the health and environmental applications. These issues are modelled using the GIS based optimization algorithm in this research. In addition, this algorithm helps to enable IoT based system and smart cities, because it is possible to provide ubiquitous health and environment services at any time and at any place with maximizing coverage and lifetime.

[92] Luo, G., Zhao, H., & Song, C. (2008, November). Convergence analysis of a dynamic discrete PSO algorithm. In 2008 First International Conference on Intelligent Networks and Intelligent Systems (pp. 89-92). IEEE.

[93] AL-Samarrie, A. K., Alyasiri, H., & AL-Nakkash, A. H. (2016). Proposed Multi-Stage PSO Scheme for LTE Network Planning and Operation. International Journal of Applied Engineering Research, 11(20), 10199-10210.

Reviewer#2, Concern #19:

Figures 7, 8 and 9 are not well explained.

Author response & action: We appreciate the reviewer’s comment. These figures have been already explained in the “4-1- Convergence Rate” and “4-2- Constancy Repeatability” sections. However, the following information have been added modified in the revised version of the manuscript.

It reads (Page 16, paragraph 2):  Figure 6 shows the global fitness values of both algorithms for 20 different executions for the WSN deployment with optimizing the coverage and the lifetime. In the figure, the results of each execution for different iterations are shown with a line. In each iteration of the BA and PSO algorithms, the positions of the sensors were randomly changed (Eq. 14 and 15) with consideration of the connectivity constraint and creating the MST graph between the sensor nodes and then the fitness value was calculated by computing the network coverage and lifetime (Eq. 3, 4, and 9). The global fitness of each iteration is the best fitness value that was obtained within the population size of the algorithms. The global fitness has changed from 0.627 to 0.755 in 100 iterations of BA while it has changed from 0.625 to 0.817 in 70 iterations of the PSO algorithm (in the best iteration). Based on the figure, both algorithms have a higher convergence rate in the first iterations and in the subsequent runs. The convergence rate is reduced and continued smoothly until global fitness is found. However, the convergence rate of the BA faster than that of the PSO algorithm in the first ten iterations because of using neighborhood searching around elite and better bees by onlooker bees in the BA algorithm. In general, the convergence rate of PSO is smoother than that of BA in all the iterations. The PSO algorithm is converged in the first 30 iterations. The reason for this issue could be small changes in the particles in each iteration. In addition, this issue shows that the PSO algorithm has better performance than BA in dealing with local optimums.

It reads (Page 17, paragraph 1): The average of the best fitness values for each iteration obtained by 10 different executions are shown in Figure 7. The figure shows, in general, the convergence of the PSO is better than that of BA and thus is better performance for network deployment for environmental and health sensors. The maximum average of fitness obtained 0.77 using the PSO algorithm and 0.72 using BA while the minimum average values of the two algorithms obtained approximately the same 0.64. In the 20 first iterations, BA found better values and in the other iterations PSO found better values. In other words, the PSO algorithm in the 20 first iterations converges to lower values of the fitness function than BA, while in the other iterations it convergence to much higher values; the best fitness value found by the PSO algorithm in the integration is around 70, but was not found by BA. These results indicate that the PSO algorithm approximately 5% on average gained better positions of the sensors regarding the trade-off between the network coverage and lifetime than BA.

It reads (page 17, paragraph 2):  In this study, it was calculated by executing both algorithms 20 times. Figure 8 shows the best result of the sequential executions of the algorithms. According to the figure, the changes in the best fitness values for the PSO algorithm are greater than those of BA in the 20 executions. These changes are from 0.702 to 0.752 for BA and from 0.713 to 0.817 for the PSO algorithm. To more accurately calculate the constancy, the standard deviation of the results of the sequential executions is obtained as follows:

Reviewer#2, Concern #20:

Both algorithms run 10 times, is it enough to do a proper statistical analysis?

Author response & action: We thank the reviewer’s comment. To calculate stability and repeatability of the multi-objective algorithms, based on the literature, the results of 10 executions (such as Saeidian et al., 2018; Fang et al., 2017; Župerl et al., 2007) or five executions (such as Masoumi et al., 2019; Niyomubyeyi et al., 2019; Masoomi et al., 2013) are compared. However, based on the reviewer’s comment, we have updated our results for 20 executions. The explanation of each figure has been updated as Concern#19.

  1. Saeidian, B., Mesgari, M., Pradhan, B., & Ghodousi, M. (2018). Optimized Location-Allocation of Earthquake Relief Centers Using PSO and ACO, Complemented by GIS, Clustering, and TOPSIS. ISPRS International Journal of Geo-Information, 7(8), 292.

Fang, Z., Li, L., Li, B., Zhu, J., Li, Q., & Xiong, S. (2017). An artificial bee colony-based multi-objective route planning algorithm for use in pedestrian navigation at night. International Journal of Geographical Information Science, 31(10), 2020-2044.

Župerl, U., Čuš, F., & Gečevska, V. (2007). Optimization of the characteristic parameters in milling using the pso evaluation technique. Strojniški vestnik-Journal of Mechanical Engineering, 53(6), 354-368.

Masoumi, Z., Van Genderen, J., & Sadeghi Niaraki, A. (2019). An improved ant colony optimization-based algorithm for user-centric multi-objective path planning for ubiquitous environments. Geocarto International, 1-18.

Niyomubyeyi, O., Pilesjö, P., & Mansourian, A. (2019). Evacuation planning optimization based on a multi-objective artificial bee colony algorithm. ISPRS International Journal of Geo-Information, 8(3), 110.

Masoomi, Z., Mesgari, M. S., & Hamrah, M. (2013). Allocation of urban land uses by Multi-Objective Particle Swarm Optimization algorithm. International Journal of Geographical Information Science, 27(3), 542-566.

Reviewer#2, Concern #21:

The authors say they perform a GIS analysis, where?,

do they know what a GIS analysis is?

Author response & action: We appreciate the reviewer’s comment. The following information have been added to the revised version of the manuscript. It reads (Page 8, Paragraph 1):

By assuming that the existing sensors can rotate 0–360° horizontally [66], each sensor coverage and total network coverage are calculated by spatial analyses. To this end, first, a point geometry layer is created using the locations of the sensors in which a point represents a sensor node. Then, the buffer polygons around the points are created to display the coverage area of the sensor in the sensing distance using buffer analysis. After that, the overlapped area between two polygons is removed from using Intersection and Union analysis. Buffer, Intersection, and Union are the spatial analyses that are performed on different types of geometry in GIS [87].

Kumar, D., Singh, R. B., & Kaur, R. (2019). Spatial Data Analysis. In Spatial Information Technology for Sustainable Development Goals (pp. 101-113). Springer, Cham.

Reviewer#2, Concern #22:

The theoretical part of the article has potential, but it must be presented much better, abstract, introduction, methodology and experimental results must be redone or greatly improved.

Other menor review is that throughout the paper, the authors have to avoid personal opinions and non-formal phrases, such as “we believe that”, “for example”, “the data needed”, etc.

It would be necessary to completely redo the paper to present it properly.

Author response & action: We appreciate the reviewer’s comment. According to the previous reviewers’ comments, we have improved the abstract, introduction, methodology, discussion, and experimental results. In addition, we have removed non-informal phrases in the revised version of the manuscript.

Round 2

Reviewer 1 Report

There are some mistypings in the text, for example:

"MST algorithm is a routing precool that minimizes WSN energy consumption"

precool should probably be protocol. 

It is still not quite clear if (and how) the information from the GIS system is used in order to affect the communication range of WSN.  Authors have added information concerning sensing range, however, from the energy consumption point of view, the communication range should be considered as well.

At line 331 it is not clear what do authors to with "number of receiving". 

Author Response

We appreciate the reviewer for his/her valuable time for reviewing our manuscript for the second round and providing us useful comments; they have certainly improved our article.

Reviewer#1, Concern # 1:

There are some mistypings in the text, for example:

"MST algorithm is a routing precool that minimizes WSN energy consumption"

precool should probably be protocol.

Author response & action: Thank you for pointing this typo out. The mentioned typo has been corrected in the revised version of the manuscript. We have also checked the whole manuscript.

MST algorithm is a routing protocol that minimizes WSN energy consumption with minimum total network lengths.

Reviewer#1, Concern # 2:

It is still not quite clear if (and how) the information from the GIS system is used in order to affect the communication range of WSN.  Authors have added information concerning sensing range, however, from the energy consumption point of view, the communication range should be considered as well.

Author response & action: Actually we have considered both communication and sensing ranges in the WSN deployment. To make it clear, following information have been added to the revised version of the manuscript.

It reads (page 6, paragraph 1): Therefore, the main issue of the WSN deployment is to find the optimal locations of the sensors by considering both the sensing and communication ranges. In addition, to find optimal locations, the connectivity constraint is very important so that the sensors that are not located in the communication range cannot be connected and to make a network.

It reads (page 10, paragraph 2):  Each bee, representing one sensor network (Figure 4), is generated by UDG with n vertices at random position. Each vertex represents a sensor node, a resource, in the WSN. The vertices that are not located in the communication range of the sensors are not connected to each other in the graph. The connectivity of the generated random graph is checked by using its adjacency matrix (G) if and only if [64].

Reviewer#1, Concern # 3:

At line 331 it is not clear what do authors to with "number of receiving".

Author response & action: We appreciate the reviewer’s comment. Based on the reviewer’s comment the following information have been modified in the revised version of the manuscript.

It reads (page 9, paragraph 2): The number of receiving is used to calculate the total energy consumptions from the viewpoint of receiving messages by Equation 12 based on the [47] because the sensors consume energy power to receive each unit message. The number of reviving messages is calculated using the number of inbound links of each sensor node.

Reviewer 2 Report

The authors have made the recommendations that I told them in the first review and, in my opinion, the paper is ready for publication.

Author Response

We appreciate the reviewer for his/her valuable time for reviewing our manuscript for the second round and providing us useful comments; they have certainly improved our article.